# Plasmepsin X activates the PCRCR complex of *Plasmodium falciparum* by processing PfRh5 for erythrocyte invasion

Tony Triglia[1], Stephen W. Scally [1,2], Benjamin A. Seager[1,2], Michał Pasternak[1,2], Laura F. Dagley [1,2] & Alan F. Cowman [1,2] ✉

*Plasmodium falciparum* causes the most severe form of malaria in humans. The protozoan parasite develops within erythrocytes to mature schizonts, that contain more than 16 merozoites, which egress and invade fresh erythrocytes. The aspartic protease plasmepsin X (PMX), processes proteins and proteases essential for merozoite egress from the schizont and invasion of the host erythrocyte, including the leading vaccine candidate PfRh5. PfRh5 is anchored to the merozoite surface through a 5-membered complex (PCRCR), consisting of Plasmodium thrombospondin-related apical merozoite protein, cysteine-rich small secreted protein, Rh5-interacting protein and cysteine-rich protective antigen. Here, we show that PCRCR is processed by PMX in micronemes to remove the N-terminal prodomain of PhRh5 and this activates the function of the complex unmasking a form that can bind basigin on the erythrocyte membrane and mediate merozoite invasion. The ability to activate PCRCR at a specific time in merozoite invasion most likely masks potential deleterious effects of its function until they are required. These results provide an important understanding of the essential role of PMX and the fine regulation of PCRCR function in *P. falciparum* biology.

Malaria is a major disease of humans responsible for more than 640,000 deaths each year[1]. *P. falciparum* causes the most severe form of the disease responsible for most of the mortality as well as considerable morbidity. Human infection by *P. falciparum* is characterised by a liver phase that ends with the release of liver merozoites from hepatocytes, and an asexual cycle, responsible for all symptoms of malaria (reviewed in ref. 2). The blood stage life cycle involves invasion of erythrocytes by merozoite forms that develop within the host cell through trophozoite and schizont stages.

Erythrocyte invasion by merozoites is a complex multi-step process (reviewed in ref. 3). The pre-invasion phase can be divided into four steps consisting of (a) initial interaction between the merozoite and erythrocyte, (b) reorientation by membrane wrapping[4], (c) discharge of apical organelle proteins and (d) tight junction formation and invasion[5,6]. As the merozoite invades the erythrocyte, it is

surrounded by the parasitophorous vacuolar membrane, a compartment within which asexual intraerythrocytic development occurs and a process that takes approximately 48 hours for *P. falciparum* and results in 16 to 32 merozoites contained within the schizont.

Once the developing merozoites are mature, a calcium-dependent rounding of the parasitophorous vacuole membrane occurs[7] and vacuolar contents enter the erythrocyte cytoplasm[8]. Plasmepsin X (PMX) inhibitors such as CWHM-117[9], 49c[10], WM4 and WM382[11,12] and the protein kinase G (PKG) inhibitor Compound 1 (C1)[13], block parasitophorous vacuole membrane rupture, so parasites remain at the rounded stage. Parasitophorous vacuole membrane rupture is the culmination of an increase in cyclic GMP (cGMP), which activates PKG[14], causing release of a subtilisin-like protease 1 (SUB1) from an apical organelle called the exoneme[15]. Calcium-dependent protein kinase 5 (CDPK5) may play a role in egress through discharge

[1]The Walter and Eliza Hall Institute of Medical Research, Parkville, VIC 3052, Australia. [2]University of Melbourne, Melbourne, VIC 3010, Australia. ✉e-mail: cowman@wehi.edu.au

of a subset of the apical organelle, the microneme[16]. The aspartic protease PMX, activates SUB1 within the exoneme after which it is released into the parasitophorous vacuole[9–11]. Here, SUB1 processes serine-rich antigens (SERAs), merozoite surface antigens (MSPs) and other proteins, causing the parasitophorous vacuole membrane and eventually the erythrocyte membrane to disintegrate[15,17]. In the final stages, the erythrocyte membrane collapses onto the parasite, curls outwards and buckles, before an explosive release of merozoites[18–20].

PMX activates a second member of the subtilisin family, SUB2, in the micronemes and processes many other microneme and rhoptry neck proteins[9–11]. Since these proteins originate in the apical organelles and often bind specific erythrocyte receptors, they are also known as invasins or adhesins. There are two main families of adhesins involved in merozoite invasion: *P. falciparum* reticulocyte-binding homologues (PfRhs) and the erythrocyte-binding like proteins (EBLs) (reviewed in ref. 3). These are large type 1 integral membrane proteins with short cytoplasmic tails and in *P. falciparum* comprise seven members: PfRh1, PfRh2a, PfRh2b, PfRh4, EBA140, EBA175 and EBA181. The tails are implicated in signalling for a subsequent step in the invasion process, such as discharge of rhoptry neck proteins[21]. PMX also proteolytically processes the essential PfRh protein, PfRh5[11]. PfRh5 lacks a transmembrane region, binds basigin on the erythrocyte membrane[22], and is a component of a 5-membered complex (called PCRCR) that includes PTRAMP, CSS, PfRipr and CyRPA[23].

Here we show that cleavage of PfRh5 by the aspartic protease PMX is essential whilst processing of PfRipr is not required for function of the PCRCR complex in merozoite invasion of human erythrocytes. These processing events occur in the micronemes which provides the means to direct the contents into the neck of the rhoptries and onto the merozoite surface. We show that removal of the PfRh5 prodomain by PMX activates the function of the PCRCR complex for binding to basigin on the erythrocyte membrane, an essential step in the invasion process. Lastly, we provide evidence that the PCRCR complex is pre-assembled in the microneme, before organellar discharge at the rhoptry neck. These results provide an understanding of the role of PMX and its importance in regulating PCRCR function during merozoite invasion by *P. falciparum* merozoites.

## Results

### Proteolytic processing of PfRh5 by PMX is essential

PfRh5 (PF3D7_0424100) is ~64 kDa and processed to a ~50 kDa protein[24] by the aspartic protease PMX releasing a prodomain of ~14 kDa[11]. The amino-acid recognition sequence for cleavage by PMX is NFLQ situated towards the N-terminus of the protein[11]. To determine if this PMX processing event was essential and to understand its role in the function of PfRh5, the endogenous *pfrh5* gene was mutated using CRISPR-cas9, to introduce mutations that would block processing of the encoded protein and simultaneously tag it with HA epitopes (Fig. 1a). Five transgenic parasites were constructed that included a control that expressed the NFLQ wild-type sequence. To block PMX cleavage we made constructs that when integrated would mutate the P2' Q to NFLA and both the P2/P2' residues (N and Q) to alanine (AFLA) (Fig. 1a, b). Two other constructs were designed to mutate one of the hydrophobic residues in addition to the Q residue (NFAA and NALA). It was expected that mutation of the PMX processing site of PfRh5 may be essential, however, surprisingly, all constructs integrated, and viable transfected parasites were obtained. The ability to obtain transgenic *P. falciparum* parasites that expressed mutant PfRh5 proteins that were expected to block processing at that site suggested either PMX cleavage was not required for PfRh5 function or there were alternative cleavage sites.

To distinguish these possibilities, proteins from each mutant parasite line were analysed (Fig. 1c). The NFLQ parasites expressing wild-type PfRh5 displayed unprocessed p64 and PMX-processed p50 products. While the NFLA and AFLA mutant parasites showed some

cleavage at the NFLQ site, both NFAA and NALA mutants showed only trace amounts of cleavage at the NFLQ site (Fig. 1c). This indicated that the P1/P1' hydrophobic residues flanking the PfRh5 cleavage site are critical for PMX cleavage (Fig. 1d). The total amount of processing (p53/p54 + p50) in the parasites ranged from ~30% to ~11% in the wild-type NFLQ and the mutant NALA parasite respectively (Fig. 1c).

The appearance of p54 in NFLA/AFLA and p54/p53 processed forms of PfRh5 in NFAA/NALA mutant parasites, suggested alternative processing sites could be utilised when the usual PMX recognition sequence was mutated (Fig. 1c). To investigate the nature of the protease(s) responsible for the alternative cleavages, a processing inhibition assay (PIA) with the PMX (WM4) and dual PMIX/PMX inhibitor (WM382)[11] was conducted on the NFAA parasite (Fig. 1e). Under control conditions, schizont rupture occurs normally and p54/p53 products are shed into the supernatant and retained in the merozoite pellet. In the presence of WM4 and WM382, schizont rupture was blocked and p54/p53 not shed and only minimally observed in merozoites. The amount of p54/p53 in the WM4 condition, showed <2% processed product, confirming that both products arise from PMX cleavage (Fig. 1e). Based on their sizes which are 3 and 4 kDa larger than the preferred cleavage at NFLQ, p53 was predicted to cleave at LLNE and p54 at VFNQ (Fig. 1f) both of which have elements of a PMX consensus sequence (Fig. 1b). These results show that when the PMX processing site was unavailable the parasite can utilise alternative sites, although less efficiently, and that this was sufficient for parasite survival, however, it also suggests that processing of PfRh5 at NFLQ to remove the N-terminus was likely essential.

To detect processing of PfRh5 in transgenic *P. falciparum* at a more sensitive level a parasite line expressing an HA-tag N-terminal to the NFLQ PMX processing sequence (Rh5PD-CHA) was constructed (Fig. 1g). Additionally, as PMX is localised to exonemes and micronemes[9], we investigated the effect of C1, a compound that blocks discharge from both organelles in *P. falciparum*[14], on processing of PfRh5. Proteins from supernatant and merozoite fractions of parasites expressing HA-tagged PfRh5 (Rh5PD-CHA) grown (+/−) C1 were assayed (Fig. 1g, h). The N-terminal prodomain of PfRh5 (Rh5PD) (p17) was detected in control supernatant and faintly in control merozoites, while C1-treated merozoites showed p17, p54/p53 and the HA-tagged p7/p6 Rh5PD products arising from alternative cleavage (Fig. 1h). This suggests that alternative processing of PfRh5 can occur in the wild-type protein but was only detected when microneme and exoneme discharge was blocked. The increased processing in C1 versus control merozoites, was most likely because C1-treatment traps PfRh5 and PMX in the micronemes, causing a higher level of processing for NFLQ and at the alternative cleavage sites.

### The function of the N-terminal domain of PfRh5 is essential

To investigate the essentiality of the N-terminal domain further, a complementation system was devised whereby mutant forms of the PfRh5 protein were expressed when integrated at the *p230p* locus in a DiCre *P. falciparum* line that expressed the endogenous protein under rapamycin inducible control (Fig. S1). The system was validated by expression of FLAG-tagged wild-type PfRh5 protein in the inducible knockout parasite (Rh5iKO), which complemented the loss of PfRh5 expression following rapamycin treatment (Fig. 2a, b).

The transgenic parasite expressing mutant PfRh5-NAAQ, which showed PMX processing only at alternative sites, gave an intermediate level of complementation suggesting there was sufficient PMX processing to allow normal merozoite invasion and growth (Fig. 2c, d). Both preferred and alternative PfRh5 cleavage sites were mutated in the Rh5-tmut parasite, however, some alternative cleavage was evident (Fig. 2e), resulting in some parasite viability as confirmed by the intermediate complementation observed (Fig. 2f). This data showed that while alternative PfRh5 PMX processing sites were sufficient for parasite viability it was clear there were also substantial fitness costs.

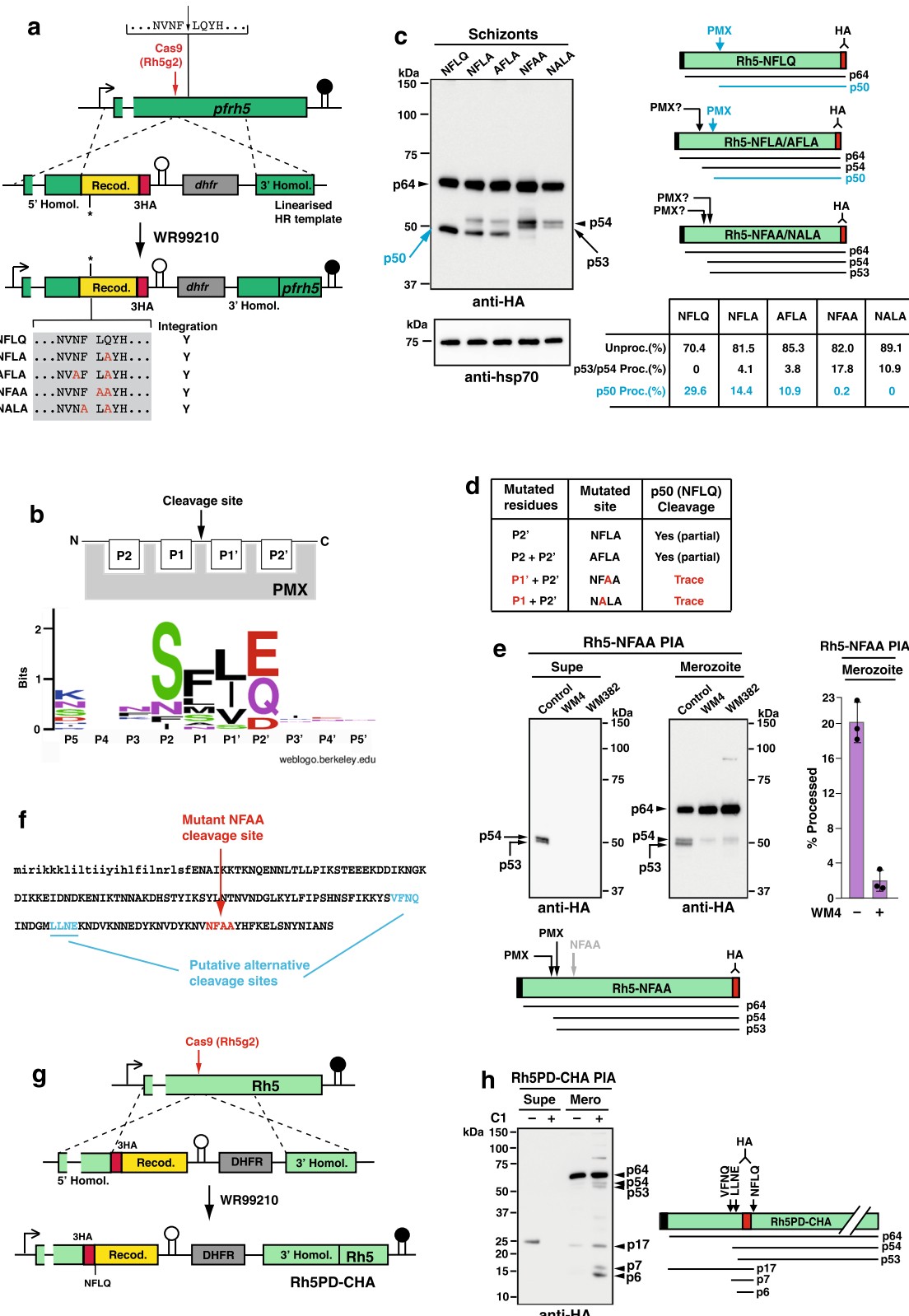

To determine if the N-terminal prodomain (PD) of PfRh5 was required for parasite growth, we performed multiple transfections of a construct that would delete the prodomain region in the endogenous *PfRh5* locus. These attempts failed to produce viable transfectants (Fig. S2), so we investigated the ability of the Rh5iKO parasite to be complemented with the expression of Rh5ΔPD. While there was good expression of Rh5ΔPD, which lacked the prodomain, there was no

complementation of parasite growth (Fig. 2g, h), confirming it was essential for parasite viability. To test if the lack of complementation was due to incorrect trafficking of the Rh5ΔPD protein, transgenic *P. falciparum* that expressed both endogenous PfRh5-HA as well as either PfRh5-HA/Rh5ΔPDnG or PfRh5-HA/Rh5nG (full-length) were constructed (Fig. S3). These proteins co-localised with the endogenous PfRh5-HA suggesting that the N-terminal prodomain of PfRh5 was not

**Fig. 1 | Mutation of PMX cleavage site in PfRh5 reveals alternative processing. a** Construction of *P. falciparum* parasites expressing mutant PfRh5 at the PMX cleavage recognition sequence. Wild-type cleavage is shown with the mutant forms listed in the grey box. Cas9 cut site in *pfrh5* is shown. *Pfrh5* genes containing *wt* or mutant PMX cleavage sites were HA-tagged. Right-pointing arrow is the endogenous *pfrh5* promoter. Black stalk depicts endogenous *pfrh5* terminator. White stalk depicts *P. berghei dhfr-ts* terminator. Selection for WR99210-resistant parasites encoded by human *dhfr*. The five transfected genes integrated into the genome (Y = Yes). **b** Schematic of PfRh5 cleavage site residues[51] and updated sequence logo (weblogo.berkeley.edu) for PMX cleavage sites. **c** Mutation of PfRh5 cleavage site identifies alternative processing sites. Immunoblots were probed with anti-HA mAb and anti-hsp70 for protein quantitation. Right panels show schematics of predicted unprocessed protein (p64), predicted PMX-cleaved product (p50) and observed cleavage fragments in the mutant parasites (p54 and p53). Cleavage at NFLQ site generating p50 highlighted in blue. The mean values for unprocessed/processed band intensities are shown. Band intensities for PfRh5 unprocessed/processed were calculated from the five parasite lines in three independent experiments. **d** Effect of mutations in endogenous NFLQ motif on cleavage by PMX. Mutation of either the P1' or P1 residue block NFLQ cleavage and generation of the p50 product. **e** Processing Inhibition Assays (PIA) with Rh5-NFAA parasites, using PMX-specific inhibitor WM4 and the PMIX/PMX dual inhibitor WM382 probed with anti-HA mAb. Lower panel shows schematic of observed products. Band intensities for PfRh5 unprocessed/processed were calculated from Control and WM4 merozoites for three independent experiments. Error bars represent standard deviation. **f** First 160 amino acids of PfRh5 showing putative alternative cleavage sites and mutated NFAA sequence. Lower-case letters are the signal sequence. **g** Construction of *P. falciparum* parasites with C-terminally HA-tagged prodomain (Rh5PD-CHA). HA-tag was placed N-terminal to the 'NFLQ' PMX cleavage site. **h** PIA with Rh5PD-CHA parasites using Compound 1 (C1) and probed with anti-HA mAb. Schematic shows expected and observed proteins. Putative non-preferred (alternative) cleavage sites (VFNQ & LLNE) and the known PMX site (NFLQ) are shown. The p17 product is the HA-tagged prodomain (PD), p7 product the HA-tagged 'VFNQ-to-NFLQ' protein and p6 the 'LLNE-to-NFLQ' protein. Source data are provided as a source data file.

---

required for correct subcellular localisation (Fig. S4). Taken together these results suggest the N-terminus of PfRh5 and its processing by PMX was essential for *P. falciparum* merozoite invasion and growth.

## The N-terminal prodomain of PfRh5 was not required for trafficking to the apical end of the merozoite

Previously, it has been shown that PfRh5 processing occurs in schizonts prior to egress, since E64-treated parasites display a PMX-processed form[11]. To understand more fully the trafficking of PfRh5 to the apical organelles of merozoites and the processing of this protein by PMX we first determined if this cleavage occurred co-translationally or in a post-Golgi compartment. The *P. falciparum* line expressing HA-tagged PfRh5 was treated with brefeldin A (BFA) and this substantially inhibited processing by PMX, indicating that this event occurred post-Golgi (Fig. S5).

We next determined if other proteins interacting with the N-terminus of PfRh5 were required for PMX processing and function (Fig. 3). WM382 was used to block PMX processing of PfRh5 and pull-down experiments were performed to identify any potential interacting proteins (Fig. S6). The five proteins of the PCRCR complex (PTRAMP (PF3D7_1218000), CSS (PF3D7_1404700), PfRipr (PF3D7_0323400), CyRPA (PF3D7_0423800), PfRh5 were significantly enriched (Fig. 3a), confirming the formation of this complex[23]. The next four most significant proteins co-precipitated were Sel1 (PF3D7_0204100), P113 (PF3D7_1420700), GRP170 (PF3D7_1344200) and 10TM (PF3D7_1208100). Sel1 and 10TM are conserved *Plasmodium* proteins of unknown function. P113 has been described and shown to interact with PfRh5 at high affinity[25], although the functional significance of this interaction has not been clear[26]. GRP170 has also been described previously[27].

To determine if these proteins are required for PfRh5 function, trafficking, or processing, we constructed parasites in which the specific gene for each could be inducibly deleted. Growth assays in the presence or absence of rapamycin (Rapa) showed that P113 and Sel1 were dispensable, however, GRP170 and 10TM were essential (Fig. 3c, g, j, m). While protein knockdown varied from ~60% (GRP170) to 100% (Sel1 and 10TM), there was essentially no effect on PfRh5 processing by PMX. These results showed that P113 (Fig. 3d), GRP170 (Fig. 3h), Sel1 (Fig. 3k) and 10TM (Fig. 3n) functions were not required for PMX processing of PfRh5 (Fig. 3e). The GRP170-iKO arrested at the late schizont stage in the first cycle (Fig. 3o), while the 10TM-iKO arrested at ring stage of the second cycle (Fig. 3p). Demonstration that P113 was not required for *P. falciparum* merozoite invasion or growth was consistent with it not playing a role in the function of PfRh5[26] and also that the other proteins detected in these pull downs with PfRh5 were false positives.

## PMX processing of PfRipr is not essential

PfRipr is part of the PCRCR complex with PfRh5 and it is essential for merozoite invasion[23,28] and processed by PMX[11]. While PMX processing of PfRh5 appears to be essential we sought to determine if this proteolytic processing of PfRipr was also required (Fig. S7a). Two transgenic *P. falciparum* lines were constructed that expressed either HA-tagged wild-type (Ripr-SMLE) or a mutated PMX processing recognition sequence (Ripr-SAAE) forms of PfRipr (Fig. S7b, c). Ripr-SMLE was processed normally, however, the mutant Ripr-SAAE was not, and only full-length protein was detected (Fig. S7c). Mutation of the PMX processing site for PfRipr had no effect on parasite growth and viability. Therefore, PMX processing of PfRipr was not a required event for the essential function of this protein in merozoite invasion[11,28].

## PCRCR proteins transit through micronemes where they are proteolytically processed by PMX

Previously, we showed the PTRAMP-CSS-PfRipr complex was pre-formed within the merozoite before surface relocation during erythrocyte invasion[23]. Consequently, we sought to determine the point at which PCRCR was PMX-processed and formed as a functional complex at the apical end of the merozoite surface.

To investigate if PCRCR complex components transit through the micronemes before they reach the rhoptry neck, parasites were treated with C1, a compound that inhibits discharge of this organelle and exonemes[14]. PfRh5 was blocked from being released into the supernatant and PMX processing of the protein in the merozoite (p46) was increased (Fig. 4a, Fig. S8a, b). Similarly, PMX processing of PTRAMP in C1-treated parasites also results in an increased accumulation of processed protein (p36) and they were not released into the supernatant (Fig. 4b). The release of PfRipr into the supernatant was also blocked by C1 and increased PMX processing in the merozoite (p64) was observed (Fig. 4c). CSS and CyRPA are not processed by PMX or any other protease, however, C1 also prevented the release of both proteins into the supernatant and increased their accumulation (Fig. 4d, e). Taken together, this data suggests that PfRh5, PfRipr, CyRPA, PTRAMP and CSS transit through the micronemes where PMX processing occurs on their pathway to the merozoite surface. It also suggests that C1-treatment results in a longer occupancy in micronemes and consequently increased PMX protease activity and accumulation of microneme proteins.

The subcellular localisation of PMX has previously been shown to be the micronemes and exonemes[9] and demonstration that C1 blocks release of this protein into the supernatant was consistent with this result (Fig. 4f, Fig. S8c). Additionally, increased autocatalytic processing of PMX was observed for C1-treated parasites consistent with increased occupancy of micronemes and autocatalytic activity. This was in comparison to PMIX, which is located in the rhoptries[9], and was

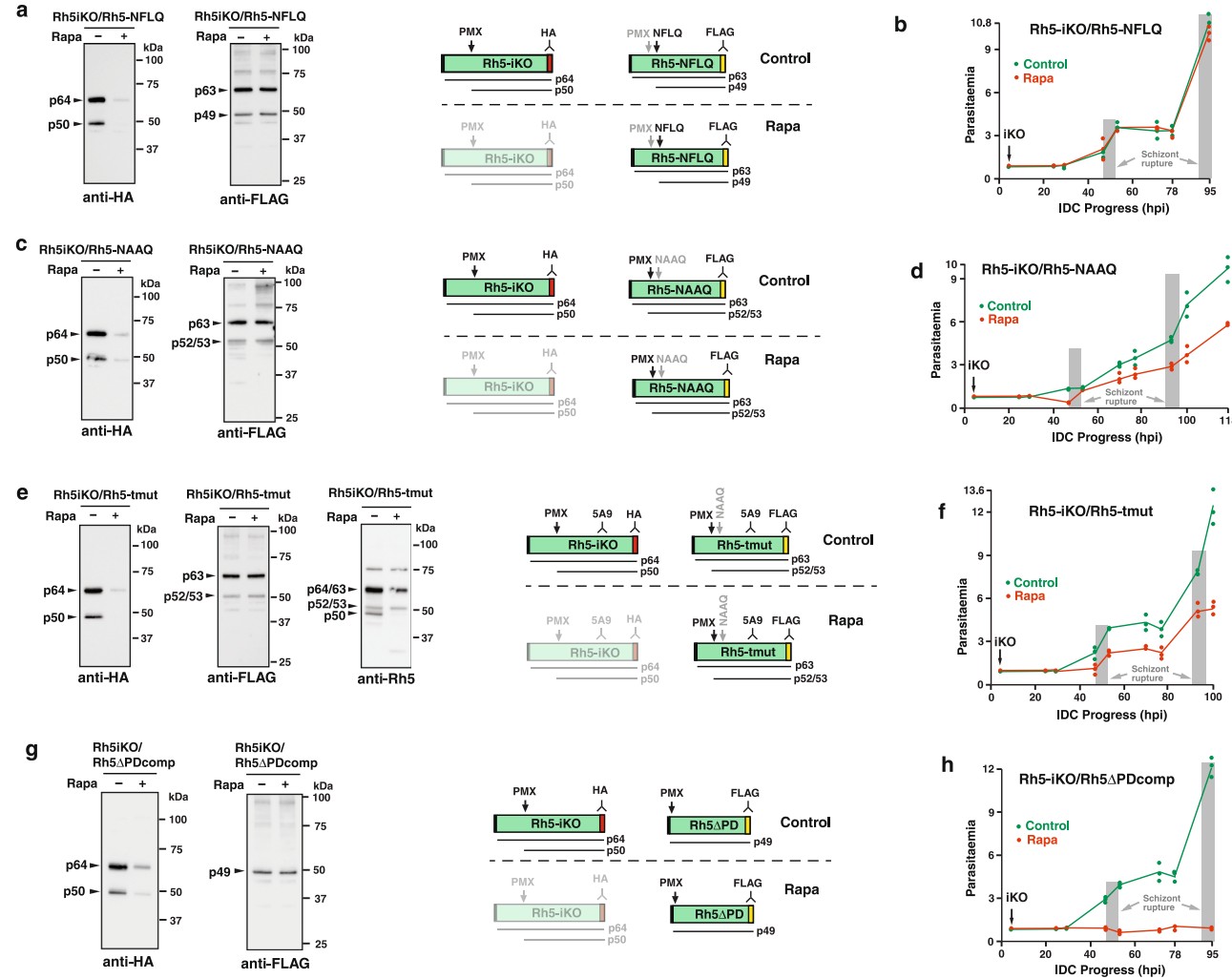

**Fig. 2 | PfRh5 N-terminal domain and processing by PMX are essential. a** Rh5-NFLQ parasite were grown+/− Rapa and probed with anti-HA and anti-FLAG mAbs. Right schematics show expected proteins in Control and Rapa conditions. Rh5-NFLQ proteins (uncleaved p63 and cleaved p49) detected with FLAG mAbs and unprocessed p64 and processed p50 PfRh5 proteins detected with HA mAbs. Protein knockdown signified by dimmed schematic. **b** Parasitaemia following+/− Rapa-treatment at 4 hr post-invasion (hpi) for Rh5iKO/Rh5-NFLQ parasites. Mean of three technical replicates+/− SD and representative of two experiments. **c** Rh5iKO/Rh5-NAAQ parasite grown+/− Rapa probed with anti-HA and anti-FLAG mAbs. Schematics show expected proteins in Control and Rapa conditions. Rh5-NAAQ proteins (uncleaved p63) and cleaved at alternative sites (p53/p52) are detected with FLAG mAbs and the unprocessed p64 and processed p50 PfRh5 proteins detected with HA mAbs. Knockdown of these proteins signified by a dimmed schematic. **d** Parasitaemia following+/− Rapa-treatment at 4 hr post-invasion (hpi) for the Rh5iKO/Rh5-NAAQ parasite. Mean of three technical replicates+/− SD and representative of two experiments. **e** Rh5iKO/Rh5-tmut parasite grown/− Rapa were probed with anti-HA, anti-FLAG and PfRh5 mAbs. Shown are schematics for expected proteins under Control and Rapa conditions. Rh5-tmut proteins (uncleaved p63 and cleaved p53/p54) detected with FLAG mAbs and the unprocessed p64 and processed p50 Rh5 proteins detected with HA mAbs. PfRh5 mAb detects both Rh5iKO and Rh5-tmut proteins to confirm the p53/54 bands detected with FLAG mAbs. Knockdown of proteins under Rapa conditions signified by dimmed schematic. **f** Parasitaemia following+/− Rapa-treatment at 4 hr post-invasion (hpi) for the Rh5iKO/Rh5-tmut parasite. Mean of three technical replicates+/− SD and representative of two experiments. **g** Rh5iKO/Rh5ΔPDcomp parasites grown+/− Rapa probed with anti-HA and anti-FLAG mAbs. On right are schematics for expected proteins in Control and Rapa conditions. Rh5ΔPD protein (p49) is detected with FLAG mAbs and unprocessed p64 & processed p50 Rh5 proteins are detected with HA mAbs. Protein knockdown signified by dimmed schematic. **h** Parasitaemia following+/− Rapa-treatment at 4 hr post-invasion (hpi) for Rh5iKO/Rh5ΔPDcomp parasite. Mean of three technical replicates+/− Standard Deviation (SD) and representative of two experiments. Source data are provided as a source data file.

retained in the merozoite pellet with and without C1-treatment because the rhoptries are not discharged until merozoite penetration and parasitophorous vacuole formation (Fig. S8d) (reviewed in ref. 3). These results show that autocatalytically active PMX was in the micronemes and explains the increased accumulation of processed polypeptides for PfRh5, PTRAMP, PfRipr and PMX itself.

To compare the results obtained for PCRCR and PMX following treatment of parasites with C1 we analysed known microneme proteins that included AMA1 and EBA140 (Fig. 4h, i). Both proteins showed similar results in which release of protein into the supernatant was blocked by C1 and an increased accumulation of proteins in the merozoite fraction was observed. These results support the conclusion

that PCRCR components are in the micronemes where they are processed by PMX before being released into the neck of the rhoptries.

PfRh5 and the other PfRh protein family members have previously been shown to have a subcellular localisation at the rhoptry neck in *P. falciparum* merozoites[29]. Our results suggest that PfRh5 was trafficked to the micronemes where it was processed by PMX before being released into the neck of the rhoptries with other PCRCR components. To test if other members of the PfRh family also transit through micronemes where they would be processed by PMX[11] we tested the effect of C1-treatment on PfRh2a/b (Fig. 4j). C1-treatment blocked the release of PMX-processed PfRh2a/b and there was an increased accumulation of the processed form of this protein. Therefore, our results

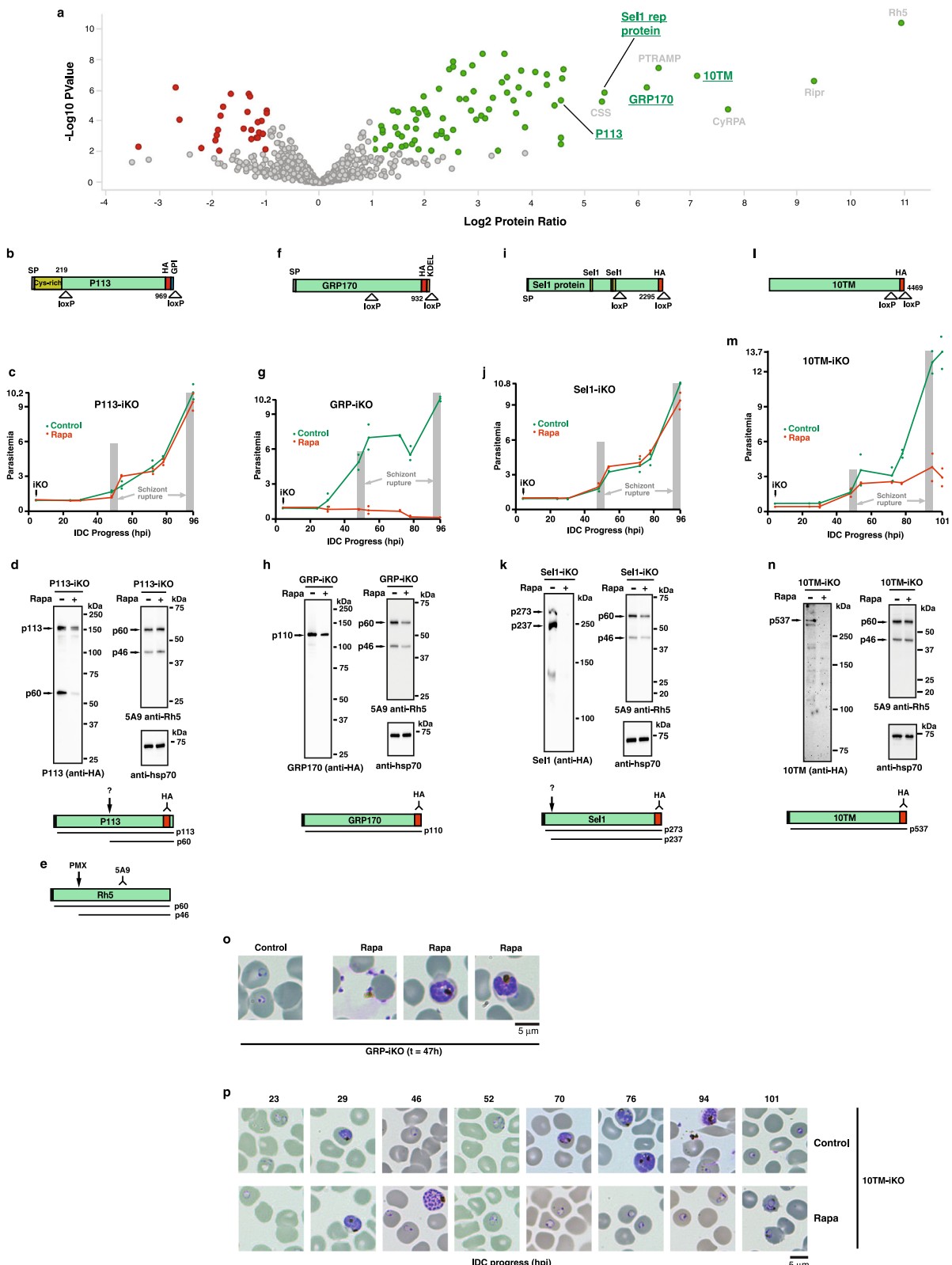

predict that PfRh proteins are trafficked through the micronemes where they are proteolytically processed by PMX before being discharged at the rhoptry neck.

**PMX inhibitors block microneme discharge of PMX and PfRh5**

The inhibitor C1 blocks cGMP-dependent protein kinase G (PKG) regulated discharge of micronemes and exonemes from *P. falciparum* merozoites by inhibiting degradation of the parasitophorous vacuole membrane and erythrocyte membrane[7,14]. To determine if PMX inhibitors had a similar phenotypic effect on discharge of microneme contents we tested the ability of WM4 to inhibit release of PMX and PfRh5 from merozoites into the supernatant. C1 blocked the release of autocatalytically processed and unprocessed PMX and PfRh5 into supernatants whereas WM4 inhibits PMX protease activity but also inhibits the release of the unprocessed forms into the supernatant (Fig. 4k). This suggests that PMX plays a role in processing and

**Fig. 3 | Post-Golgi trafficking of PfRh5 does not require interaction with a PfRh5-specific protein. a** *P. falciparum* parasites with a N-terminally HA-tagged prodomain (Rh5PD-NHA) were generated (Fig. S6). Synchronised 3D7 and Rh5PD-NHA were grown with WM382 (to prevent Rh5PD cleavage) to the schizont stage, uninfected erythrocytes lysed with saponin, then parasites treated (+/−) the DSP crosslinker. Twelve samples representing four in triplicate for Mass Spectrometry (MS) were prepared. The samples were: 3D7, 3D7 + DSP, Rh5PD-NHA and Rh5PD-NHA + DSP. Proteins were immunoprecipitated with agarose-bound anti-HA antibodies, trypsin digested and analysed by MS. The results of the comparison between Rh5PD-NHA + DSP and 3D7 + DSP are shown. All components of the PCRCR complex[23] were detected (ie. PTRAMP, CSS, Ripr, CyRPA, Rh5). In addition, four other proteins with high significance were found: 10TM (PF3D7_1208100), GRP170 (PF3D7_1344200), Sel1 repeat protein (PF3D7_0204100) and P113 (PF3D7_1420700). **b**, **f**, **i**, **l** Representations for the four proteins targeted for inducible knockout using the DiCre system are shown. For P113 and GRP170, the HA-tag was placed N-terminal to the GPI anchor or the 'KDEL' ER retention signal, respectively. For Sel1 protein and 10TM, the HA-tag was placed at the C-terminus of the protein. The position of the loxP sites shows how much of each protein is removed in the presence of Rapa. **c**, **g**, **j**, **m** Constructs for each gene were transfected into the Pfs47-DiCre line to allow inducible knockout in the presence of Rapa. Parasitaemia following +/− Rapa-treatment at 4 hr post-invasion (hpi) for the four parasites is shown. Shown are data points for one experiment. **d**, **e**, **h**, **k**, **n** Proteins from synchronised schizonts from each parasite grown with or without Rapa, were blotted, then probed with anti-HA, anti-Rh5 and anti-hsp70 Abs. Below the immunoblots are schematics for the expected proteins to be produced upon probing with the anti-HA and anti-Rh5 mAb. Note that a processing event was detected for the P113 and Sel1 proteins. **o** Giemsa-stained images of the GRP170-iKO parasite at *t* = 47 hr, in the control and Rapa conditions. **p** Giemsa-stained images of the 10TM-iKO parasite through 2 asexual life cycles, under both control and Rapa conditions. Source data are provided as a source data file.

activating proteins required for rupture of both the parasitophorous vacuole membrane and the parasite-infected erythrocyte membrane.

### Functional PCRCR requires proteolytic processing of PfRh5 by PMX

To further investigate the function of the N-terminal domain of PfRh5 that is proteolytically removed by PMX both full-length (Rh5$_{unproc}$) and the equivalent processed form lacking the N-terminal domain (Rh5$_{proc}$) were expressed and purified from insect cells[23]. The Rh5$_{proc}$ and Rh5$_{unproc}$ forms have previously been shown to bind basigin on the erythrocyte membrane surface as well as in solution[23,30].

To ensure both Rh5$_{proc}$ and Rh5$_{unproc}$ were functional, biolayer interferometry was used to test their ability to interact with components of the PCRCR complex[23,30]. It was determined that CyRPA, which interacts directly with both PfRh5 and PfRipr[30], bound both Rh5$_{proc}$ and Rh5$_{unproc}$ at approximately the same dissociation constants (K$_d$) suggesting they interacted at close to equal affinity (Fig. 5a). Additionally, the ability of Rh5$_{proc}$ and Rh5$_{unproc}$ to participate in formation of the PCRCR complex was compared by firstly, testing their ability to bind the preformed PfRipr-CyRPA complex. Both proteins bound to PfRipr-CyRPA equally well and secondly, to the cysteine-linked PTRAMP-CSS hetero-dimer (Fig. 5b)[23]. These showed that both Rh5$_{proc}$ and Rh5$_{unproc}$ were able to form the PCRCR complex equally well consistent with their normal function.

To determine if Rh5$_{proc}$ and Rh5$_{unproc}$ were able to bind to basigin on the surface of human erythrocytes, we used fluorescence-activated cell sorting (FACS) and detected binding using anti-PfRh5 antibodies. This showed that Rh5$_{proc}$ bound to human erythrocytes at levels like that described previously[23,30], however, the full-length Rh5$_{unproc}$ did not show any apparent binding (Fig. 5c). PCRCR, in which either the Rh5$_{proc}$ or the Rh5$_{unproc}$ forms were incorporated in the complex, were then tested for their ability to bind human erythrocytes and PfRh5 detected using specific antibodies. The PCRCR-Rh5$_{proc}$ complex bound efficiently, however, no binding was detected for PCRCR-Rh5$_{unproc}$. This data suggests that the N-terminal domain of PfRh5 masks the basigin binding domain to ensure the PCRCR complex is inactive until its function is required during merozoite invasion of human erythrocytes (Fig. 5d).

### Discussion

The aspartic protease PMX has been shown to be responsible for processing of multiple proteins required for *P. falciparum* growth, egress and merozoite invasion, however, the functional significance of these events has been unknown[9–11]. PfRh5 is an essential protein that functions in the PCRCR complex for binding to basigin on the merozoite surface[22,23,30] and processing of this protein by PMX is an essential step that removes the prodomain to uncover the basigin binding site thus activating the function of this complex. This suggests that the PCRCR complex is in an inactive state until its function is required at merozoite invasion. The reasons for this are currently unknown, however, it is possible that early presence of an active complex may have deleterious effects on the merozoite such as insertion into membranes as occurs for PfRh5 and PfRipr during merozoite invasion[30].

Mutation of the PMX cleavage recognition sequence of PfRh5 revealed alternative processing sites but this also showed that this processing event was essential. These alternative processing sites can be cleaved in the endogenous wild-type PfRh5 by PMX, however, it was at an inefficient level although sufficient for viability, albeit with major fitness costs. WM4 and WM382 are potent inhibitors of PMX and more advanced compounds of this chemical class are being developed as potential novel antimalarials and an understanding of their mode of action is important. Inhibition of PfRh5 processing by PMX is one of the important targets of these compounds and this results in inhibition of merozoite invasion. However, WM382 and WM4 not only block processing of PMX but also many other protein substrates and also activation of proteases required for merozoite egress, invasion and parasite development and this would explain the observed high potency of these compounds in inhibiting growth of *P. falciparum*[11]. WM382 has been shown to be a potent inhibitor of *P. falciparum* at liver, blood and transmission stages and further development is in progress of a clinical candidate that can be tested in human clinical trials as an antimalarial drug.

The *P. falciparum* egress programme begins ~9 min before merozoites exit the erythrocyte (Fig. 6a). Both PMX and SUB1 reside in the exonemes[9], and following their fusion and discharge, SUB1, released into the parasitophorous vacuole space, begins the process of parasitophorous vacuole breakdown (Fig. 6b). Micronemes containing EBA and AMA1 proteins fuse at the rhoptry neck before the invasion but SUB2, which also has a microneme subcellular localisation, is discharged on the surface of the merozoite proteins during active invasion[31]. This supports the hypothesis that there are different forms of micronemes as has been suggested previously for both *P. falciparum*[16,32,33] and *Toxoplasma gondii*[34]. Our data adds support to this idea and results from fractionation of C1-treated merozoites suggests there are different types of micronemes that contain distinct cargo (Fig. 6a–e). We hypothesise that PCRCR, EBAs, PfRhs, AMA1 and PMX are stored in micronemes type 1 and 2 before egress. Consistent with this is the observation that PMX processing of micronemal contents does not result in full processing of all proteins and increases markedly when microneme fusion was blocked by C1 (Fig. 6f). This suggests that these proteins are separated from PMX at some stages and that either there are likely two early microneme subsets, one of which contains PMX (Fig. 6g), or alternatively these micronemes have subcompartments (Fig. 6h). Previous evidence for microneme subsets in *P. falciparum*[16,32,33] have suggested that EBA175 and AMA1 are in separate micronemes. The proposed microneme subsets are in addition to the exonemes[15] and mononemes[35] previously defined.

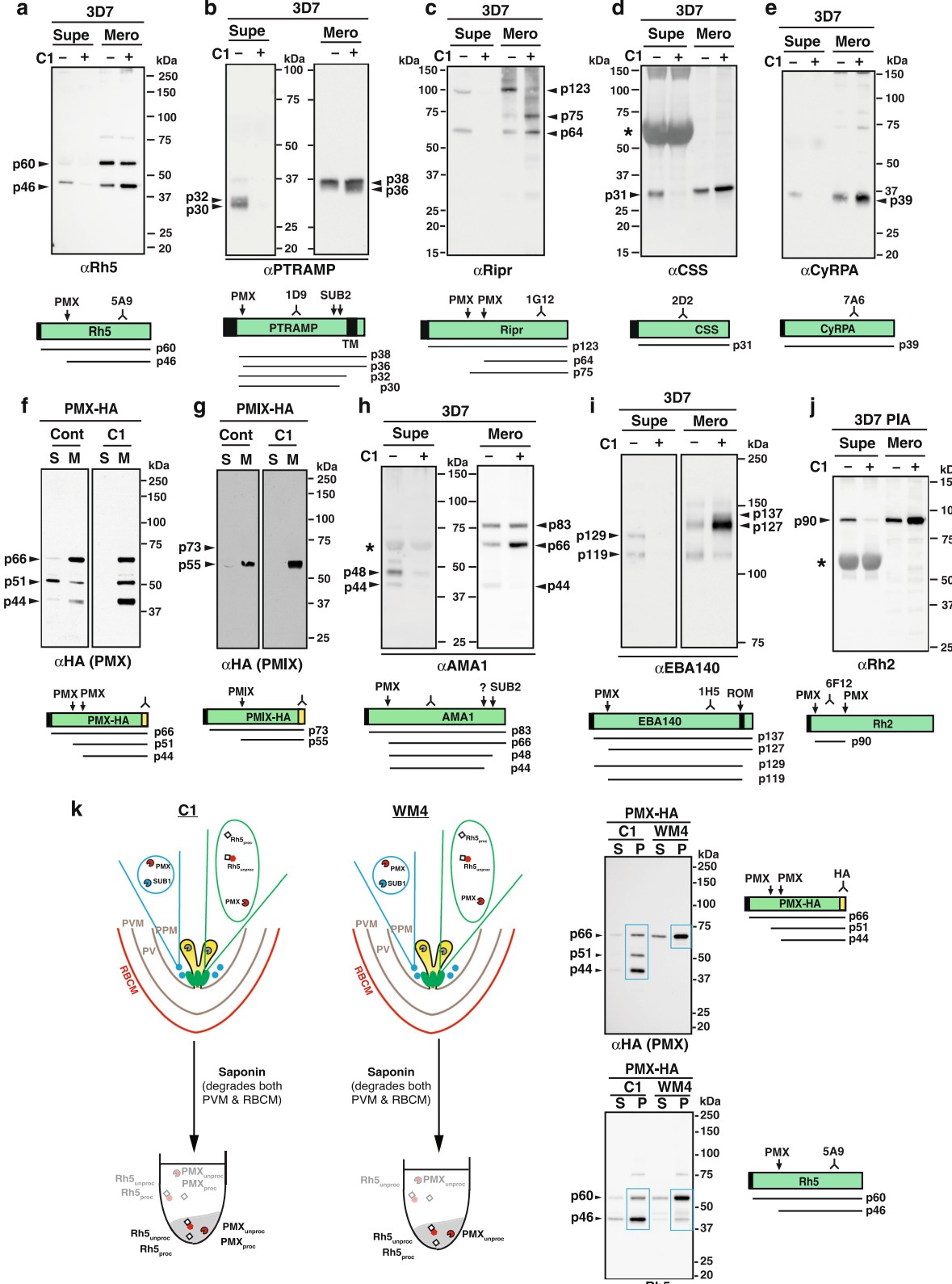

The compartmentalisation of proteins destined for processing by PMX and relocation to the merozoite surface during invasion provides a mechanism to time specific processing events and release of protein as required for the finely controlled process of merozoite invasion.

Recent work suggested that PfRh5 was PMX-processed in the parasitophorous vacuole[36], however, our results show that PMX was discharged from both exonemes and microneme type 1 and 2 to the

parasitophorous vacuolar space (Fig. 6a–c) as a processed form. Increased PMX processing of proteins after C1-treatment was not compatible with PfRh5 cleavage occurring in the parasitophorous vacuole. Additionally, the pH optimum of PMX is 5.5 suggesting that the micronemes are acidic to activate autocatalysis of this protease[11] and once released into the parasitophorous vacuole its activity would be expected to markedly decrease due to a higher pH.

**Fig. 4 | The PCRCR complex is processed by PMX and formed in micronemes or a microneme subset.** Immunoblots on proteins found in different compartments of the merozoite: microneme, rhoptry neck, rhoptry body and exoneme. Processing Inhibition Assays were set up with either 3D7, PMX-HA and PMIX-HA parasites. The derivation of PMIX-HA, and PMX-HA[11] parasites has been described. All assays included C1. Supernatant (S or supe) and merozoite (M or mero) fractions are shown. Below each blot is a diagram of the expected products under Control and C1 conditions. **a** Immunoblot of the rhoptry neck protein, PfRh5. **b** Immunoblot of the microneme protein, PTRAMP. **c** Immunoblot of the microneme protein, PfRipr. **d**, **e** Immunoblots of the microneme proteins, PfCSS and PfCyRPA. **f** Immunoblot of the exoneme and microneme protease, PMX. **g** Immunoblot of the rhoptry body protein, PMIX. **h**, **i** Immunoblots of the microneme proteins, AMA1 and EBA140. **j** Immunoblot of the rhoptry neck protein, PfRh2. All proteins were separated under reducing conditions, except the PfRipr proteins. The approximate location of the mAb or pAb used for the immunoblots is shown in the schematic for each protein, together with the calculated size of the unprocessed or processed product in kDa. The asterisk (*) indicates a cross-reaction of some Abs with albumin in supernatant fractions. **k** WM4 blocks exoneme and microneme discharge. The merozoite fractions from a Processing inhibition assay on PMX-HA parasites treated with either C1 or the PMX inhibitor, WM4, were saponin-lysed. Proteins from equal volumes of the saponin supernatant (S) and saponin pellet (P) were separated by SDS-PAGE and probed with either HA or Rh5 mAbs. The illustrations of C1 and WM4-treated merozoites are identical, since the data shows that WM4 just like C1, blocks exoneme and microneme discharge. The blue boxes delineate the proteins detected in the saponin pellets (i.e., remaining within the merozoites) under both treatments. Source data are provided as a source data file.

The subcellular localisation of PfRh5 and other members of the PfRh protein family has been defined previously as the rhoptry neck which has led to the conclusion that proteins originating in different organelles come together at the apical tip of egressing merozoites[24,28]. Both PfRh5, PfRipr and CyRPA do not show full co-localisation in merozoites which suggested there are pools that have not yet formed the RCR or the PCRCR complex and presumably reflect their trafficking from the Golgi to the apical organelles as monomers rather than a complex[28]. Transit of the PCRCR components through micronemes suggests that the complex is formed at this point and PMX processing occurring subsequently, is consistent with the ability of the unprocessed PfRh5 protein to form the PCRCR complex. The presence of pools of unprocessed PMX, PfRh5, and PfRipr suggests that the protease is not activated until a late stage just before invasion and this would cause a cascade of processing events including activation of SUB1 and SUB2 (Fig. 6d, e, Fig. S9). The existence of different forms of micronemes with distinct protein cargoes would provide a means to separate proteins and provide conditions to store inactive proteases such as PMX until they are required.

Previously, PfRh5 has been expressed in mammalian cells as a full-length fusion protein in a pentameric platform and this enabled the detection and identification of basigin as the receptor to which it bound on the erythrocyte surface[22]. This result would be inconsistent with our demonstration that full-length PfRh5 did not bind basigin because the 15 kDa prodomain blocked the binding site. However, a subsequent publication showed that the PfRh5 protein with a ~25 kDa Cd4 tag, migrates at 70 kDa suggesting that PfRh5 had been processed[37]. The PfRh5 prodomain that is removed by PMX is predicted to be highly disordered and readily cleaved by unknown proteases both in insect[38], Schneider 2[39] and mammalian heterologous expression systems. Therefore, assays testing the ability of PfRh5 to bind erythrocytes and basigin directly would have very likely used the processed PfRh5 that lacked the N-terminal domain consistent with our results that the N-terminal prodomain blocks basigin binding.

The presence of a highly disordered prodomain at the N-terminus of PfRh5 that can be processed by different proteases in heterologous systems also suggests it can interact with different proteins. The identification of P113, GRP170, Sel1, and 10TM is consistent with this, however, the analysis of conditional knockouts of the corresponding genes encoding these proteins showed that they have no effect on PMX processing of PfRh5 suggesting that these interactions are not functionally relevant and false positives. The ER protein GRP170 has previously been analysed and plays a role in trafficking of proteins from the ER, however, it does not appear to be required for PfRh5 localisation to micronemes because it does not affect PMX processing which occurs at this later stage[27]. P113 has been shown to bind to the prodomain of PfRh5 and has been hypothesised as the protein that anchors PfRh5 to the merozoite membrane[25]. However, other studies have shown that P113 is localised to the parasitophorous vacuole membrane where it is associated with the Plasmodium translocon of exported proteins (PTEX) and plays a role in the integrity of this compartment[26]. Our results showing that P113 is pulled down with the PCRCR complex was consistent with its interaction with the prodomain of PfRh5. However, our demonstration that removing the expression of P113 has no effect on merozoite invasion or growth of *P. falciparum* shows that this protein plays no role in the PCRCR complex and that its interaction with PfRh5 is functionally irrelevant. Our inability to find a specific trafficking protein that affects PfRh5 processing suggests that it may be trafficked directly from the Golgi to its apical organelle through a common escorter protein[40], such as PfSortilin whose proposed role is to carry cargo destined for the apical organelles[41]. How the *P. falciparum* Golgi ensures specificity for specific apical organelles or compartments remains unknown.

In conclusion, this work has identified a role for PMX in processing PfRh5 and this activates the PCRCR complex for its essential role in merozoite invasion. This provides an increased understanding of the roles of PMX, PfRh5, and the fine regulation required for PCRCR and its function in *P. falciparum* biology.

## Methods
### Ethics statement
Use of human blood and serum was approved by the Walter and Eliza Hall Institute of Medical Research Human Ethics committee under approval number 19-05VIC-13.

**Parasite culture.** *P. falciparum* asexual blood stage parasites were grown in O + erythrocyte (Australian Red Cross, South Melbourne, Australia) at 4% haematocrit in Roswell Park Memorial Institute (RPMI) 1640 medium supplemented with 26 mM 4-(2-hydroxyethyl) piperazine-1-ethanesulfonic acid (HEPES), 50 μg/ml hypoxanthine, 20 μg/ml gentamicin, 0.2% NaHCO$_3$, 0.5% Albumax II$^{TM}$ (Gibco). Cultures were incubated at 37°C in a mix of 94% N$_2$, 1% O$_2$ and 5% CO$_2$.

**Parasite lines expressing HA-tagged proteins.** Transgenic parasite lines were made using CRISPR-Cas9 (Table S1) as previously described[11]. Guide oligos designed to induce a double-stranded break in the corresponding genomic positions were cloned using InFusion into pUF1-Cas9G (Table S2). The strategy involved generation of a guide plasmid and a plasmid that replaces the endogenous target gene with a tagged version (the homology repair or HR plasmid). The HR plasmids assembled in a modified p1.2 plasmid encoding WR99210 resistance, were made in three steps, with 5' and 3' flanks (~500 bp upstream or downstream from the guide sequence) amplified from 3D7 genomic DNA and a codon-optimised target gene sequence (Genscript) cloned downstream of the 5' flank. Linearised HR plasmid (50 μg) and circular guide plasmid (100 μg) were transfected together into synchronised 3D7 schizonts suspended in 100 μl of P3 primary cell solution. Programme FP158 with the Amaxa P3 primary cell 4D Nucleofector X Kit L (Lonza) was used. Parasites with an integrated drug-resistance cassette were selected and maintained on 2.5 nM WR99210. Plasmids used and generated in this study are described in Table S3.

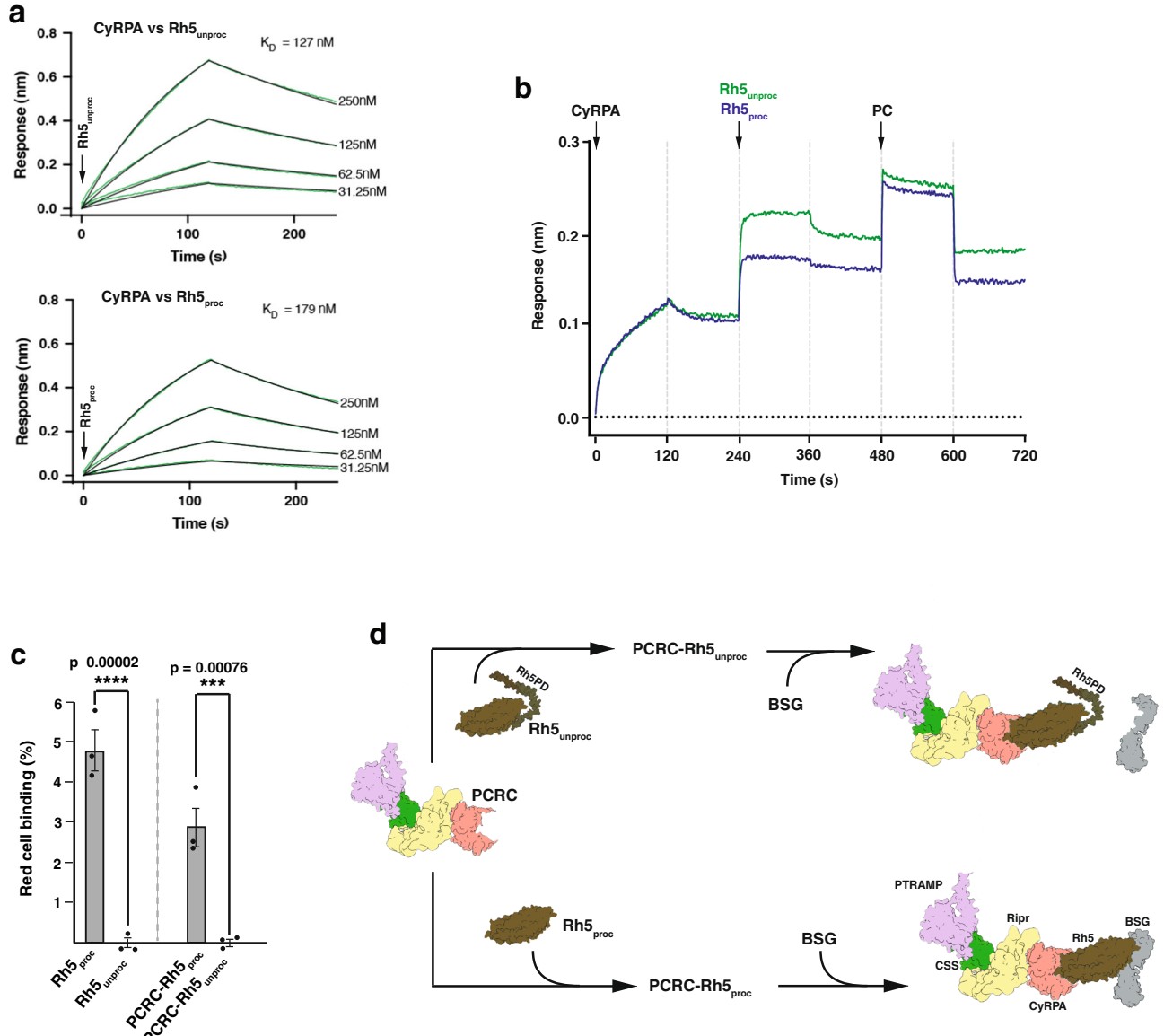

**Fig. 5 | The role of the N-terminal prodomain of PfRh5 in the PCRCR complex.**
**a** Binding of CyRPA to either Rh5$_{unproc}$ or Rh5$_{proc}$ was determined by Biolayer Interferometry (BLI). Biotinylated CyRPA was pre-added to a Streptavidin biosensor, before dipping into twofold dilutions of Rh5$_{unproc}$ and Rh5$_{proc}$ at $t = 0$ s. Binding occurred for 120 s followed by dissociation for a further 120 s. The $K_D$ values were calculated using Octet Data Analysis v11.0 software. **b** Binding of Rh5$_{unproc}$ and Rh5$_{proc}$ to the RCR complex by BLI. CyRPA was added at $t = 0$ to biotinylated Ripr immobilised on a Streptavidin biosensor for 120 s before allowing dissociation for a further 120 s. Either Rh5$_{unproc}$ or Rh5$_{proc}$ were added to Ripr-CyRPA and allowed to bind, then dissociate, before PTRAMP-CSS (PC) was added at

$t = 480$ s. Binding and dissociation were allowed to occur until $t = 720$ s. **c** Binding of Rh5$_{unproc}$ or Rh5$_{proc}$ and PCRC-Rh5$_{unproc}$ or PCRC-Rh5$_{proc}$ to erythrocytes measured by flow-cytometric analysis. The assay was done three times and means (+/−) standard error of the mean (SEM) is shown. One-way ANOVA with Sidak's multiple comparisons test was used to calculate $p$ values. **d** Representation of the binding results in **c** using known structures of CSS (PDB ID: 7UNZ), PfRipr, CyRPA, PfRh5 (PDB ID: 6MPV), basigin (BSG) (PDB ID: 3B5H) and the AlphaFold Monomer v2.0 structures[52] of the PTRAMP ectodomain (Uniprot: Q8I5M8) and Rh5PD (Uni-Prot: Q8IFM5). Structures were assembled in ChimeraX (https://www.rbvi.ucsf.edu/chimerax). Source data are provided as a source data file.

**Parasite lines with inducible gene knockouts.** Transgenic parasite lines for Sel1-iKO, P113-iKO, GRP170-iKO and 10TM-iKO were made as above (Table S1) except plasmids were transfected into the Pfs47-DiCre line[42] to enable regulated deletion of specific genes using the dimerisable Cre system. Guide oligos for InFusion cloning are detailed in Table S2. The HDR plasmids were made as for the HA-tagged parasites, except the codon-optimised sequences included a *loxP* site within an introduced *sera5* intron and a second loxP site, following the STOP codon, was part of the plasmid. For DiCre excision, synchronised schizont cultures were allowed to rupture till few ring stages were present, followed by sorbitol-synchronisation to remove schizonts,

then grown with 10 nM Rapa or DMSO, until the schizont stage of the first cycle (~46 hr). Uninfected erythrocytes were removed with saponin lysis and the remaining schizont proteins obtained by lysing schizont pellets in reducing SDS sample buffer.

**Parasite lines expressing both an inducibly expressed tagged Rh5 and Rh5 variants at the P230p locus.** Parasites that inducibly expressed PfRh5 (Rh5iKO) and that were already WR99210-resistant, were re-transfected with constructs to express four PfRh5 variants from the *P230p* locus (Rh5iKO/Rh5ΔPDcomp, Rh5iKO/Rh5-NAAQ, Rh5iKO/Rh5-tmut and Rh5iKO/Rh5-NFLQ). These constructs were

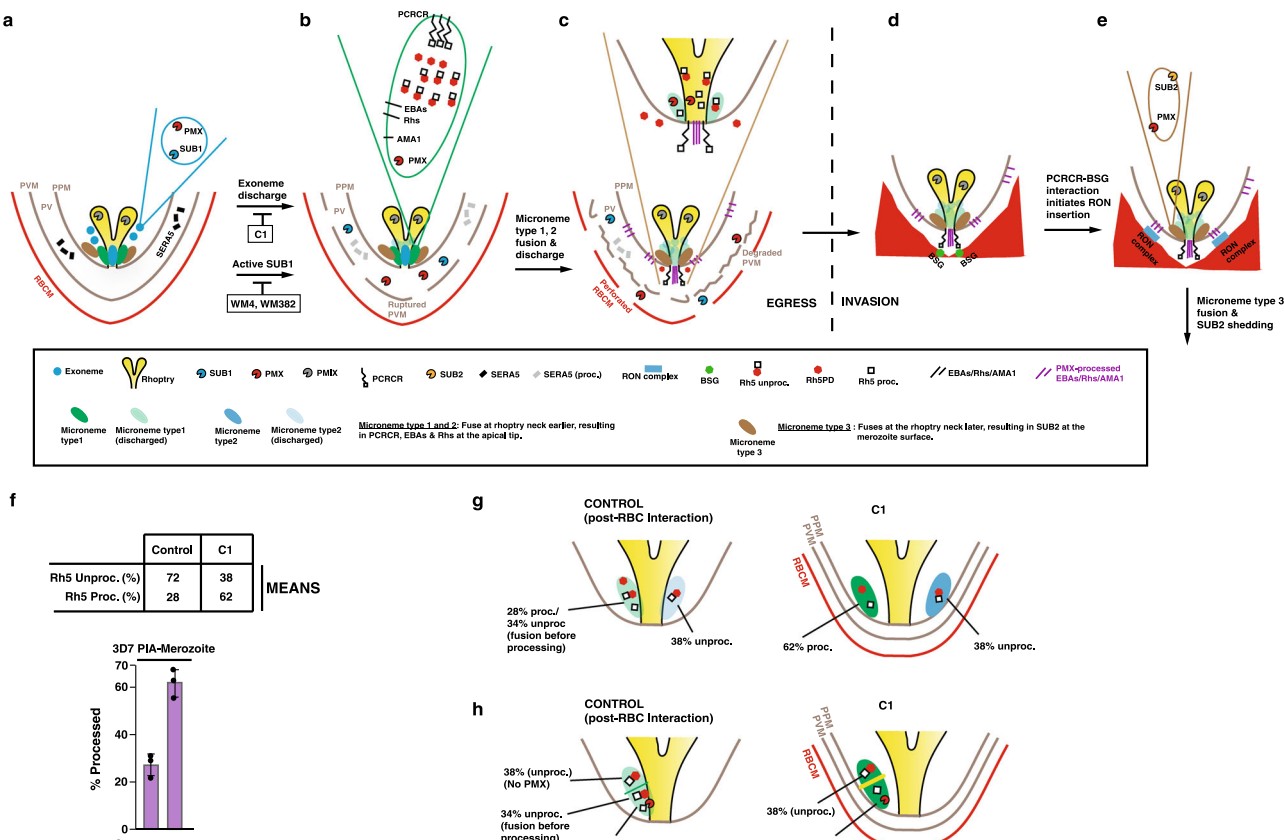

**Fig. 6 | Model of protein processing and organellar discharge during merozoite egress and invasion. a** Merozoite of a schizont bounded by parasite plasma membrane (PPM) within the PVM and erythrocyte membrane. PfSUB1 and PMX are stored in exonemes. C1 blocks exoneme and microneme discharge, while PMX inhibitor WM4 and dual PMX/PMIX inhibitor WM382 block parasitophorous vacuole membrane (PVM) and erythrocyte membrane degradation as SUB1 is inactive. **b** Following exoneme fusion and discharge, SUB1 and PMX are discharged into the parasitophorous vacuole (PV). Here, SUB1 processes several PV-resident proteins (such as SERA5) culminating in a ruptured PVM and a permeabilised erythrocyte membrane. A subclass of microneme is postulated to contain EBAs, PfRhs, AMA1, PMX and PCRCR. In this study, ~30% of PfRh5 was PMX-processed to the Rh5proc form, while 70% remains unprocessed, as depicted in the microneme contents. Rh5proc: PMX-processed PfRh5; Rh5unproc: Unprocessed PfRh5; Rh5PD; PfRh5 prodomain. **c** Following microneme fusion and discharge, the EBAs, PfRhs and PCRCR complex are translocated to the parasite membrane at the apical tip, while AMA1 initially inserts into the merozoite membrane, then spreads over the merozoite surface. Further membrane lytic events result in a degraded PVM and perforated erythrocyte membrane. **d, e** Merozoites egress from the erythrocyte ready to engage fresh erythrocyte. PfRh and EBAs are involved in early invasion steps, followed by binding of PCRCR to basigin (BSG), which culminates in release of RON complex proteins into the RBC membrane. Processed AMA1 on the merozoite surface engages with inserted RON complex, to form the tight junction for merozoite entry. We propose that SUB2 'sheddase' is contained in a second microneme subset also containing PMX, that fuses at the rhoptry neck and merozoite membrane, releasing activated SUB2 to shed proteins from the merozoite surface. **f** Proteins from the merozoite fraction of a 3D7 processing inhibition assay treated (+/−) C1, were probed with an anti-Rh5 mAb and the PfRh5 unprocessed and processed band intensities calculated. Mean values are shown for the level of PfRh5 processing, shown in the histogram below. Band intensities for PfRh5 unprocessed/processed were calculated from Control and C1-treated merozoites in three independent experiments. Error bars represent standard deviation. **g, h** Data comparing PfRh5 processing levels in C1-treated merozoites (Fig. 6f) suggests two possible models for microneme heterogeneity. The first model (**g**) posits there are two early microneme subsets (type 1 and 2), only one of which contains PMX. Under control conditions, this subset fuses at the rhoptry neck before PfRh5 has been completely PMX-processed, giving a mix of 28% processed and 34% unprocessed. The other subset has no PMX, so upon rhoptry neck fusion, contributes a further 38% unprocessed PfRh5. Under C1 conditions where there is no rhoptry fusion, PfRh5 processing in the microneme subset goes to completion at 62% and leaving 38% unprocessed. The second model (**h**) is like the microneme subset model, except that the microneme is compartmentalised into two sections, only one of which contains PMX, but both compartments contain PfRh5. Source data are provided as a source data file.

---

made in a p1.2 plasmid where the *hdhfr* gene was replaced with the *blasticidin deaminase* (*bsd*) gene (Table S1). Transfection conditions were as before, but selection was with 2.5 µg/ml blasticidin.

**Parasite lines expressing both an endogenous tagged Rh5 and Rh5 variants at the P230p locus.** Parasites that expressed a tagged PfRh5 (Rh5HA) and that were already WR99210-resistant, were re-transfected with constructs to express two Rh5 variants from the *P230p* locus (Rh5HA/Rh5ΔPDnG and Rh5HA/Rh5nG). These constructs were made in a p1.2 plasmid where the *hdhfr* gene was replaced with the blastic*idin deaminase* (*bsd*) gene (Table S1). Transfection conditions were as before, but selection was with 2.5 µg/ml blasticidin.

**Parasite growth assay.** Transgenic ring-stage parasites, in which a specific gene could be conditionally deleted, were synchronised to a parasitaemia between 0.5 and 0.8%, and grown in the presence of Rapa or DMSO (as a control) in triplicate wells. Parasite smears were taken for Giemsa staining at approximately 22, 29, 46, 52, 70, 76, and 94 hr post-invasion. One thousand cells were counted at each time point to determine the parasitaemia.

**Processing inhibition assay (PIA).** Parasites used to analyse the effect of various drugs on proteins in merozoite and supernatant fractions, were treated as previously described[11]. Synchronised late trophozoite/early schizont cultures to which protease inhibitors (WM4 or WM382)

or the PKG inhibitor C1 (pyrrole 4-[2-(4-fluorophenyl)−5-(1-methylpi-peridine-4-yl)−1H-pyrrol-3-yl] pyridine) had already been added, were passed over LS magnetic columns (Miltenyi Biotech) to remove uninfected erythrocytes. The PMX inhibitor WM4 was used at 40 nM, while the dual PMX and PMIX inhibitor WM382, was used at 2.5 nM and C1 at 1 µM final concentration. A control dish without drug was also included. Parasites were eluted from columns with complete RPMI 1640 culture medium to which the appropriate inhibitor at the same concentration had been added. Eluted parasites were adjusted to $5 \times 10^6$ schizonts/mL and 150 ml added per well of a 96-well flat-bottomed culture dish. The assay dishes were further cultured for 16 hr and a representative well from each condition smeared for Giemsa staining, to ensure either that rupture had occurred normally (control well) or that rupture had been blocked (C1, WM4, and WM382 conditions). Parasites from each condition were centrifuged at $10,000 \times g$/10 min to separate merozoite and supernatant fractions. Proteins were extracted in reducing sample buffer.

**PIA and saponin fractionation.** PMX-HA parasites[11] were treated with C1 or WM4 as above. Uninfected erythrocytes were removed with LS columns and assays incubated overnight. Parasites were centrifuged at $10,000 \times g$/10 min to obtain the merozoite fraction, which was saponin-lysed with 300 µl 0.125% saponin. The saponin pellet was made to 300 µl reducing sample buffer. Equal volumes of both supernatant and pellet were run for Western blots.

**SDS-PAGE and immunoblotting.** Proteins from saponin-lysed schizonts or supernatant and merozoite fractions from PIA assays, were separated on 4%–12% or 3%–8% acrylamide gels (NuPAGE, Invitrogen). Separated proteins were transferred to nitrocellulose membranes by electroblotting. Blots were probed with HRP-conjugated anti-HA antibody (Roche) 1:1000 or for two-step methods, a primary antibody was followed by HRP-conjugated secondary antibody (Millipore). Bands were detected using ECL Plus Western blotting reagent (GE Healthcare) and the ChemiDoc Imaging System (Biorad). Antibodies used in this study are described in Table S4.

To ensure low MW proteins less than 20 kDa as in Fig. 1f, remained on the nitrocellulose membrane during the blocking, antibody and washing steps, the membrane was subject to a post-blot MeOH fixation protocol[43]. Briefly, after electroblotting, the membrane was placed in 50% MeOH for 30 min/4 °C, then 50% MeOH for 30 min/50 °C, before the usual Western blot procedure.

**Antibodies.** The full list of antibodies together with the dilutions used is described in Table S4. In this study, we used the following antibodies: rat mAb anti-HA (Roche 3F10); mouse mAb anti-nGreen (Chromotek 32F6); mouse mAb 1D9 anti-PfPTRAMP[23], rat mAb 2D2 anti-CSS[23], mouse mAb 7A6 anti-CyRPA[44], mouse mAbs 5A9 and 6H2 anti-PfRh5[38], mouse mAb 1H5 anti-EBA140[11], mouse mAb 6F12 anti-Rh2a/b[29], rabbit pAb anti-SERA5[45] and rabbit pAb anti-AMA1[46].

**Crosslinking and immunoprecipitation.** 3D7 and Rh5PD-NHA parasites were tightly synchronised, then left untreated or treated with 2.5 nM WM382 until they reached the late schizont stage. Schizonts were saponin-lysed to remove uninfected erythrocytes. For crosslinking protein complexes, 0.1 mM (final) of the thiol-cleavable crosslinker DSP (dithiobis (succinimidyl propionate)), was added to saponin-lysed pellets for 30 min at room temperature. An equivalent reaction without crosslinker was used as a control. The reaction was quenched with 20 mM Tris, before proteins were extracted with nine pellet volumes of TNET (1% TX100, 150 mM NaCl, 10 mM EDTA, 50 mM Tris, pH 7.4). Protein extracts were incubated with agarose-bound anti-HA antibodies (Roche) and immunoprecipitants eluted with hot 0.5% SDS at 56 °C for 5 min. Chemicals and reagents used in this study are described in Table S5.

**Trypsin digestion of HA immunoprecipitations.** Eluates of HA-captured proteins derived from each biological replicate were prepared for mass spectrometry analysis using the FASP (filter-aided sample preparation) method[47], with the following modifications. Proteins were reduced with 10 mM Tris-(2-carboxyethyl) phosphine (TCEP), alkylated with 50 mM iodoacetamide, then digested with 1 µg sequence-grade modified trypsin gold (Promega) in 50 mM $NH_4HCO_3$ and incubated overnight at 37 °C. Peptides were eluted with 50 mM $NH_4HCO_3$ in two 40 µl sequential washes and acidified in 1% formic acid (FA, final concentration).

**Mass spectrometry analysis.** The extracted peptide solutions from immunoprecipitation experiments were acidified (0.1% formic acid) and concentrated by centrifugal lyophilisation using a SpeedVac AES 1010 (Savant). For the 3D7 and Rh5PD-NHA samples, peptides were reconstituted in 80 µl 2% ACN/0.1% FA and 3 µl separated by reverse-phase chromatography on a C18 fused silica column (inner diameter 75 µm, OD 360 µm × 25 cm length, 1.6 µm C18 beads) packed into an emitter tip (IonOpticks, Australia), using a nano-flow HPLC (M-class, Waters). The HPLC was coupled to a timsTOF Pro (Bruker) equipped with a CaptiveSpray source. Peptides were loaded directly onto the column at a constant flow rate of 400 nl/min with buffer A (99.9% Milli-Q water, 0.1% FA) and eluted with a 90-min linear gradient from 2 to 34% buffer B (99.9% ACN, 0.1% FA). The timsTOF Pro was operated in PASEF mode using Compass Hystar 5.1. Settings for the 11 samples per day method were as follows: Mass Range 100 to 1700 m/z, 1/K0 Start 0.6 V s/cm² End 1.6 V s/cm², Ramp time 110.1 ms, Lock Duty Cycle to 100%, Capillary Voltage 1600V, Dry Gas 3 l/min, Dry Temp 180 °C, PASEF settings: 10 MS/MS scans (total cycle time 1.27 sec), charge range 0–5, active exclusion for 0.4 min, Scheduling Target intensity 10000, Intensity threshold 2500, CID collision energy 42 eV.

For the 3D7 and Rh5PD-NHA samples, raw files consisting of high-resolution tandem mass spectrometry spectra were processed with MaxQuant (version 1.6.17) for feature detection and protein identification using the Andromeda search engine[48]. Extracted peak lists were searched against the *P. falciparum* 3D7 database and a separate reverse decoy database to empirically assess the FDR using a strict trypsin specificity allowing up to two missed cleavages. The minimum required peptide length was set to seven amino acids. The modifications included: carbamidomethylation of Cys was set as a fixed modification, whereas N-acetylation of proteins and the oxidation of Met were set as variable modifications. The 'match between runs' option in MaxQuant was used to transfer the identifications made between runs based on matching precursors with high mass accuracy. LFQ quantification was selected, with a minimum ratio count of 2. Peptide-spectrum match (PSM) and protein identifications were filtered using a target-decoy approach at an FDR of 1%. In the main search, precursor mass tolerance was 0.006 Da and fragment mass tolerance was 40 ppm.

**Immunofluorescence and super-resolution microscopy.** For fixed imaging, synchronised schizonts were fixed with 4% Paraformaldehyde and 0.01% glutaraldehyde for 30 min, permeabilised with 0.1% TX100 in HTPBS for 30 min and incubated in blocking solution (2% BSA in PBS) for 1 hr. Primary antibodies rat anti-HA (Roche 3F10, 1:300) and mouse anti-nGreen (1:300) were used. Secondary Alexa 488/594 fluorophores were used at 1:1000 dilution. Parasites were mounted on coverslips coated with 1% poly-ethyleneimine and mounted with Vectashield containing DAPI (VectorLabs, Australia). Z-stacks of fluorescently labelled infected red blood cells were imaged with Zeiss LSM880 inverted microscope equipped with a Plan Apochromat ×63/1.4 oil objective with 405, 488, 561, and 594 nm excitations and an Airyscan detector. Colocalization between channels was measured in ImageJ2/Fiji using the JACoP plugin with Costes automatic threshold.

## Cell lines

Sf21 insect cells were cultured in Insect-XPRESS Protein-free with L-Glutamine (Lonza, 10036636) medium at 28 °C. Expi293F cells were grown in Expi293™ Expression medium (Thermofisher) at 37 °C, 8% $CO_2$, 120 RPM.

## Expression and purification of Rh5<sub>unproc</sub>, Rh5<sub>proc</sub>, CyRPA, Ripr, and PTRAMP-CSS

The gene for Rh5$_{unproc}$ (residues 26 to 526) was subcloned into pACGP67a with an N-terminal Flag tag and TEV protease cleavage site and a C-terminal C-tag. Three potential N-linked glycosylation sites Asn214, Asn284 and Asn297 were removed by mutation of Thr or Ser residues to Ala. The construct was expressed in Sf21 cells and secreted into the medium as a soluble protein. The supernatant was purified by ANTI-FLAG M2 Affinity Gel (Merck) and size exclusion chromatography (S200 Increase 10/300 GL, Cytiva). Fractions containing Rh5$_{unproc}$ were pooled and cleaved with TEV protease for 16 h at 4 °C. His-tagged TEV was removed via NiNTA Agarose resin (Qiagen) and Rh5$_{unproc}$ was further purified via another size exclusion chromatography step (S200 Increase 10/300 GL, Cytiva).

Rh5$_{proc}$ (residues 145–526), was purified as previously described[23]. Briefly, the gene for Rh5$_{proc}$ was subcloned into pACGP67a with a C-terminal C-tag. Three potential N-linked glycosylation sites Asn214, Asn284 and Asn297 were removed by mutation of Thr or Ser residues to Ala. The construct was expressed in Sf21 cells and secreted into the medium as soluble protein. The supernatant was purified by CaptureSelect C-tagXL Affinity Matrix (Thermofisher) and eluted with 20 mM Tris pH 7.5, 2 mM $MgCl_2$. The sample was further purified via size exclusion chromatography, using an S200 Increase 10/300 GL column (Cytiva).

The gene for CyRPA (residues 29 to 362) was subcloned into a modified pcDNA3.4-TOPO plasmid with an N-terminal IL-2 signal sequence and a C-terminal Flag preceded by a TEV protease cleavage site. Three potential N-linked glycosylation sites at Asn145, Asn322 and Asn338 were removed by mutation of the glycan site Thr or Ser residues to Ala. The construct was expressed via transient transfection of Expi293F cells and soluble protein was purified from the culture medium in a similar manner to Rh5$_{unproc}$ described above. In addition, a CyRPA construct with a C-terminal Avitag in place of the TEV protease cleavage site was generated and specifically biotinylated[49] for coupling to High Precision Streptavidin (SAX) biosensors in biolayer interferometry experiments.

The gene for *pfripr* (residues 20 to 1086) was subcloned into pACGP67a with an N-terminal Avitag and a C-terminal His-tag. The construct was expressed in Sf21 cells and secreted into the medium as soluble protein. The supernatant was dialysed into 20 mM Tris pH 8, 150 mM NaCl. Imidazole was added to 10 mM final concentration and PfRipr was purified by NiNTA Agarose (Qiagen) and eluted in 20 mM Tris pH 8, 150 mM NaCl, 500 mM Imidazole. The sample was further purified via size exclusion chromatography, using a S200 Increase 10/300 GL (Cytiva) and specifically biotinylated[49].

PTRAMP-CSS was generated by co-expressing PTRAMP and CSS constructs in Sf21 insect cells. The PTRAMP (42-297) construct was subcloned into pACGP67a with a potential N-linked glycosylation site at Asn195 removed by mutation of Thr197 to Ala and a C-terminal C-tag. The CSS (20-290) construct was subcloned into a modified pTRIEX2 vector with potential N-linked glycosylation sites at Asn74, Asn192, Asn234 and Asn261 removed by mutation of N-linked glycan site Ser or Thr residues to Ala, and a C-terminal Flag tag preceded by a TEV protease cleavage site. The insect cell supernatant was purified by ANTI-FLAG M2 Affinity Gel (Merck) and size exclusion chromatography (S200 Increase 10/300 GL, Cytiva). Fractions containing PTRAMP-CSS were pooled and cleaved with TEV protease for 16 h at 4 °C. His-tagged TEV was removed via NiNTA agarose resin (Qiagen),

and PTRAMP-CSS was further purified via another size exclusion chromatography (S200 Increase 10/300 GL, Cytiva).

## Biolayer interferometry studies

Biolayer interferometry experiments were conducted at 25°C to determine the affinity of Rh5 constructs for components of the PCRCR complex. For protein-protein binding kinetic studies, biotinylated CyRPA was diluted into kinetics buffer (PBS, pH 7.4, 0.1% (w/v) BSA, 0.02% (v/v) Tween-20) at 20 µg/mL and immobilised onto High Precision Streptavidin (SAX) biosensors (Sartorius). Following a 60 s baseline step, biosensors were dipped into wells containing twofold dilution series of either Rh5unproc or Rh5proc. Sensors were then dipped back into kinetics buffer to monitor the dissociation rate. To test binding of Rh5unproc and Rh5proc to the Ripr-CyRPA complex, biotinylated Ripr was diluted into kinetics buffer at 20 µg/mL and immobilised onto High Precision SAX biosensors (Sartorius). Following a 60 s baseline step, biosensors were dipped into wells containing 5 µM CyRPA. Sensors were then dipped back into kinetics buffer to monitor the dissociation rate. CyRPA's slow dissociation rate ensured it stayed mostly bound to Ripr. Biosensors were then dipped into wells containing 5 µM of either Rh5unproc or Rh5proc. Sensors were then dipped back into kinetics buffer to monitor the dissociation rate. Biosensors were then dipped into wells containing 5 µM of PTRAMP-CSS and then dipped back into kinetics buffer to again monitor the dissociation rate.

## Flow-cytometric analysis of erythrocyte binding

Erythrocyte-binding assays were performed as described[23] with the following minor changes: samples were analysed on an Attune NxT flow cytometer (Thermo Fisher Scientific) and 100,000 events were recorded.

## Statistics and reproducibility

Blots shown in Fig. 1c, h; 2a, c, e, g; 3d, h, k, n–p and Fig. S3d; S5; S6; S7c are representative of two or three independent experiments. Software used in this study is described in Table S6.

## Reporting summary

Further information on research design is available in the Nature Portfolio Reporting Summary linked to this article.

## Data availability

Antibodies, primers, parasite lines, and reagents used in this study are available in Table S1–5. The mass spectrometry proteomics data have been deposited in the ProteomeXchange Consortium via the PRIDE[50] partner repository with the dataset identifier PXD039646 with the username reviewer_pxd039646@ebi.ac.uk and password pdJIA5kW. Structures used are CSS (PDB ID: 7UNZ), PfRipr, CyRPA, PfRh5 (PDB ID: 6MPV), basigin (BSG) (PDB ID: 3B5H). Source data are provided with this paper.

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

## Acknowledgements

We thank Australian Red Cross Blood Service for blood, Walter and Eliza Hall Institute Monoclonal Laboratory for monoclonal antibodies, and Ellen Knuepfer for the Pfs47-DiCre parasite line. This work was possible through grants from The Wellcome Trust (109662/Z/15/Z (to A.F.C.), 202749/Z/16/Z (to A.F.C.)) and the National Health and Medical Research Council of Australia (NHMRC) (APP1194535 to A.F.C.) and Victorian State Government Operational Infrastructure Support grant (Institutional grant).

## Author contributions

T.T. designed, constructed, and analysed all mutant and conditional knockout parasites, wrote the original draft, and reviewed and edited the manuscript. S.W.S. expressed proteins, and performed and analysed protein-protein interaction studies. B.A.S. performed red cell binding FACS experiments. M.P. performed the imaging experiments. L.F.D. analysed mass spectrometry experiments. A.F.C. designed and interpreted experiments and reviewed and wrote the manuscript.

## Competing interests

The authors declare no competing interests.
