## [Peer Review File · Nature Communications]

Plasmepsin X activates the PCRCR complex of *Plasmodium falciparum* by processing PfRh5 for erythrocyte invasionREVIEWER COMMENTS

Reviewer #1 (Remarks to the Author):

This work by Triglia et al. provides key insights into the timing, localization, activity, and functional importance of the aspartic protease plasmepsin X (PTEX) on the formation of the PCRCR complex. The noteworthy finding of this study is that PCRCR is processed by PMX in micronemes which activates the function of the complex, allowing Pfrh5 to bind basigin, mediating downstream erythrocyte invasion. PMX processing of Pfrh5 is essential, yet PMX processing of Pfripr is not. An additional key finding was the observation that among proteins that were co-precipitated and interacting with Pfrh5, P113 and Sel1 were dispensable, whereas GRP170 and 10TM were essential. The essential functions of GRP170 and 10TM were independent of PTEX processing of Pfrh5. This study employs rigorous methodology and reveals precise mechanistic details and fundamental insights into the dynamics of essential (and non-essential) protein interactions involved in the critical process of erythrocyte invasion.

The methodology is sound and rigorous, and the findings justify the conclusions. The complementary approaches presented in Extended Data provide additional supporting evidence to further justify the conclusions.

It is rare that I review a manuscript and have so few suggestions for improvements. While I have no major points, I hope these few minor points and suggestions might marginally improve an already impressive and significant work.

Minor Points:

1. There are a few instances in the paper where essentiality is inferred from the inability to recover transfectants. Yet, the authors have successfully employed inducible knock-outs and complementation with mutants to experimentally confirm essentiality. Occasionally the emphasis is placed on the negative result. In these instances, it would be helpful to emphasize the clear lack of complementation with the mutant in the inducible system as the primary evidence and phrase the inability to recover stable mutant transfectants as supporting secondary evidence.

a. Line 197-200 – “multiple transfections of a construct that would delete the region encoding the pro-domain in the endogenous pfrh5 locus failed to produce viable transfectants. Taken together these results suggest the N-terminus of Pfrh5 and its processing by PMX was essential for P. falciparum merozoite invasion and growth.” The experiment using the DiCre inducible replacement system to prove, rather than infer essentiality, is performed in Fig 2 g,h, but the way the section is written, the emphasis is placed on the inability to recover transfectants rather than the failure to complement with the DiCre system. Perhaps by inverting the order of presentation of the data would clarify this point. The authors could state that they were unable to produce viable transfectants, and so they employed the inducible replacement approach to specifically address essentiality. The order in which the data was described for the PTEX processing mutants: Lines 122-128, followed by lines 171-186 was effective.

2. Given the essentiality of PTEX processing of Pfrh5, the promise of PTEX inhibitors as potential antimalarial is exciting. The authors allude to this in line 354-355, but it would be interesting to discuss any current or potential limitations to PTEX inhibitors as drug targets. What is the toxicity profile of the current inhibitors, bioavailability, etc.?

I applaud the authors for this excellently designed and executed study that provides fundamental insights into merozoite biology and yields deeper insights into the function of PTEX and the PCRCR complex and its role in the molecular mechanism of erythrocyte invasion.

Reviewer #2 (Remarks to the Author):

Review of paper by T. Triglia et al. submitted to Nature Medicine.

This is an outstanding paper that describes the processing of RH5 by PMX. There is a great deal of detail and methods that increase our understanding of the RH5 complex in invasion. They show that the binding to Basigin requires the cleaving of the N-terminus of RH5. They produced cleaved

and uncleaved RH5 and show that only cleaved will bind to Basigin. I would point out that the discovery of RH5 and Basigin (ref 22 Crosnier C. et al) identified the binding with intact RH5 and this worked because they used the AVEKIS system that allows identity although the interaction is low affinity.

My one concern is Fig. 6 in which they introduce a new term, preneme, a subclass of micronemes. Evidence for this is not mentioned in the results section. This organelle enters the rhoptry apical end. There is no data to substantiate the process that they describe. The minimum would be electron microscopy showing these events. Without these data, they need to say that this division of micronemes is purely speculative. More important, they need to develop these data in the results or leave it for another publication. I am recommending acceptance of the paper when corrected for Nature Medicine.

Review by Louis H. Miller

Reviewer #3 (Remarks to the Author):

A complex of proteins including PfRh5, which the authors have recently identified new members for and renamed the whole complex PCRCR, plays a key role in the invasion of human erythrocytes by Plasmodium falciparum merozoites. Several members of this complex are proteolytically cleaved during the invasion process, as are numerous other P. falciparum invasion proteins. In this manuscript, the authors use a combination of CRISPR-Cas9 engineering and protease inhibitors to explore the role that this processing plays for PCRCR location and function. The manuscript is well written, clear, contains extensive data and will undoubtedly add significantly to our understanding of this critical step during invasion. It should also be noted that in several places the authors are absolutely pushing the boundaries of technology – Figure 2 in particular is a genetic tour de force, this combination of inducible knockouts and complementation with different variants is rarely performed in P. falciparum and to be applauded. The only significant concern I have is that the hypotheses about different kinds of secretory organelles (prenemes/pronemes etc) and their roles in invasion veers a little far into the speculative, and the figures presenting them lend an air of definitiveness that is not currently backed up by the data. Other than that, a few relatively minor technical items should be addressed in a revised manuscript.

Major issues

1) Strain adaptation. The authors note that two of the PfRh5 PMX cleavage-site edited lines, NFAA and NLAA, took significantly longer to grow following transfection (lines 122-124), which matches with the more significant PfRh5 processing defect in these lines. It would be helpful to provide more details here – how much longer did they take to come back, and was there a persistent growth defect, or once the parasites were recovered, did they grow at the same rate as any other line? A formal growth rate comparison post-recovery would be helpful. However, the deeper concern with all P. falciparum transfections is that during the extended time it takes to recover lines, other genetic/epigenetic changes have emerged in addition to the targeted mutation. These could be stochastic but could also be compensatory – ie. other changes were needed to compensate for these specific PMX site mutations. RNAseq and/or whole genome sequencing of the edited lines would help provide reassurance that the phenotypes observed are not in part due to background changes in addition to the targeted mutations/or that additional changes were needed for the NFAA and NLAA mutations to be viable. Of course, this is a common feature of all P. falciparum transfections, and is almost always overlooked, so it is acknowledged that this is asking the authors to go further than almost all similar papers do. There may be good reasons why this is not required – it would be helpful though to hear that reasoning articulated.

2) Function of interacting proteins. Figure 3 is a tour de force, and almost a manuscript in itself – pull down of interacting proteins, then generating inducible knockouts of the encoding genes to assess function, is a huge amount of work. The implications for PCRCR function/PfRh5 function however are not completely clear. The authors test for an effect on PfRh5 processing in the iKO lines and see no effect, but what about PfRh5 trafficking and/or PCRCR complex formation? If none of these elements are affected, then is the inference that the interactions detected by co-immunoprecipitation are actually false positives?

3) Trafficking models. Figure 6 lays out an extended model of merozoite invasive organelles, suggesting there are two different kinds of micronemes (prenemes and postnemes) and that prenemes are further functionally divided. While developing new hypotheses to bring the field forward is of course highly commendable and useful, the fact remains that the definition of these new organelles/subtypes of micronemes does not include direct visualisation/microscopy – it is all theoretical, based on the authors' interpretation of inhibitor experiments, and is therefore highly speculative. The concern is that these terms and definitions, laid out in a very definitive way in Figure 6, will enter standard usage without solid experimental justification. Discussing the potential implications of the processing data for trafficking is reasonable enough, but Figure 6 is, in my opinion, advancing hypothesis too far into speculation.

Minor items

1) Definitiveness of interpretation of immunoblots. Much of the processing data interpretation depends on the presence/absence of bands on immunoblots, where the authors will know well different exposure times can sometimes support very different interpretations. The blots are clear and well controlled, so this is not a significant concern, but statements like "NFAA and NALA mutants were not processed at the NFLQ site" (line 133) is more absolute than Fig1c, with a faint processed band that could perhaps be lighter or darker depending on exposure, actually warrants. A simple tempering of language, and acknowledging where appropriate that it is not possible to be absolute based on immunoblot data alone is all that is required.

2) Quantification of co-localisation. The PfRh5 prodomain is stated to co-localise with full-length PfRh5, and Ext Fig 3 is compelling, but only two schizonts are shown for each co-localisation. How many schizonts were measured overall using super-resolution microscopy to make such an absolute statement about co-localisation? Being less absolute, and more quantitative (eg Rh5HA/Rh5nGreen co-localise $x\% \pm y\%$ in n schizonts, and Rh5HA/Rh5delPDnGreen co-localise $a\% \pm b\%$ in n schizonts) would be more accurate and provide more reassurance in the data.

Reviewer's**Comments:****Reviewer #1** (Remarks to the Author):

This work by Triglia et al. provides key insights into the timing, localization, activity, and functional importance of the aspartic protease plasmepsin X ('PTEX', but should be PMX) on the formation of the PCRCR complex. The noteworthy finding of this study is that PCRCR is processed by PMX in micronemes which activates the function of the complex, allowing Pfrh5 to bind basigin, mediating downstream erythrocyte invasion. PMX processing of Pfrh5 is essential, yet PMX processing of Pfripr is not. An additional key finding was the observation that among proteins that were co-precipitated and interacting with Pfrh5, P113 and Sell were dispensable, whereas GRP170 and 10TM were essential. The essential functions of GRP170 and 10TM were independent of PMX processing of Pfrh5. This study employs rigorous methodology and reveals precise mechanistic details and fundamental insights into the dynamics of essential (and non-essential) protein interactions involved in the critical process of erythrocyte invasion.

The methodology is sound and rigorous, and the findings justify the conclusions. The complementary approaches presented in Extended Data provide additional supporting evidence to further justify the conclusions.

It is rare that I review a manuscript and have so few suggestions for improvements. While I have no major points, I hope these few minor points and suggestions might marginally improve an already impressive and significant work.

General Authors response: We thank the reviewer for taking the time to read the paper and providing feedback.

Minor Points:

1. There are a few instances in the paper where essentiality is inferred from the inability to recover transfectants. Yet, the authors have successfully employed inducible knock-outs and complementation with mutants to experimentally confirm essentiality. Occasionally the emphasis is placed on the negative result. In these instances, it would be helpful to emphasize the clear lack of complementation with the mutant in the inducible system as the primary evidence and phrase the inability to recover stable mutant transfectants as supporting secondary evidence.

a. Line 197-200 – “multiple transfections of a construct that would delete the region encoding the pro-domain in the endogenous *pfrh5* locus failed to produce viable transfectants. Taken together these results suggest the N-terminus of PfRh5 and its processing by PMX was essential for *P. falciparum* merozoite invasion and growth.” The experiment using the DiCre inducible replacement system to prove, rather than infer essentiality, is performed in Fig 2 g,h, but the way the section is written, the emphasis is placed on the inability to recover transfectants rather than the failure to complement with the DiCre system. Perhaps by inverting the order of presentation of the data would clarify this point. The authors could state that they were unable to produce viable transfectants, and so they employed the inducible replacement approach to specifically address essentiality. The order in which the data was described for the PMX processing mutants: Lines 122-128, followed by lines 171-186 was effective.

Authors response: We agree that the lack of complementation is the primary evidence and the inability to obtain transfectants with the Rh5 Δ PD construct is confirming evidence, so we have altered the Results section to reflect this change, as below:

“To determine if the N-terminal prodomain (PD) of PfRh5 was required for parasite growth, we performed multiple transfections of a construct that would delete the prodomain region in the endogenous *PfRh5* locus. These attempts failed to produce viable transfectants (Extended Data Fig. 4), so we investigated the ability of the Rh5iKO parasite to be complemented with expression of Rh5 Δ PD. While there was good expression of Rh5 Δ PD, that lacked the prodomain, there was no complementation of parasite growth (Fig. 2g, h), confirming it was essential for parasite viability. To test if the lack of complementation was due to incorrect trafficking of the Rh5 Δ PD protein, transgenic *P. falciparum* that expressed both endogenous PfRh5-HA as well as either PfRh5-HA/Rh5 Δ PDnG or PfRh5-HA/Rh5nG (full-length) were constructed (Extended Data Fig. 2). These proteins co-localised with the endogenous PfRh5-HA suggesting that the N-terminal prodomain of PfRh5 was not required for correct subcellular localisation (Extended data Fig. 3). Taken together these results suggest the N-terminus of PfRh5 and its processing by PMX was essential for *P. falciparum* merozoite invasion and growth.”

2. Given the essentiality of PMX processing of PfRh5, the promise of PMX inhibitors as potential antimalarial is exciting. The authors allude to this in line 354-355, but it would be interesting to discuss any current or potential limitations to PMX inhibitors as drug targets. What is the toxicity profile of the current inhibitors, bioavailability, etc.?

Authors response: We thank the reviewer for the comments. WM382 has been further developed to a clinical candidate stage and this compound called MK-7602 will be published at a later date. However, we cannot disclose the name and properties of this clinical candidate in this manuscript. WM382 is a lead compound, however, it is also our main tool compound that has the same properties as MK-7602. Whilst WM382 has potent activity against the parasite and also *in vivo* activity for animal models it is not a drug appropriate for humans.

We have added an extra comment in the discussion as shown below to provide some more detail.

‘Mutation of the PMX cleavage recognition sequence of PfRh5 revealed alternative processing sites but this also showed that this processing event was essential. These alternative processing

sites can be cleaved in the endogenous wild-type PfRh5, however, it was at an inefficient level although sufficient for viability, albeit with major fitness costs. WM4 and WM382 are potent inhibitors of PMX and more advanced compounds of this chemical class are being developed as potential novel antimalarials and an understanding of their mode of action is important. Inhibition of PfRh5 processing by PMX is one of the important targets of these compounds and this results in inhibition of merozoite invasion, however, the presence of alternative cleavage sites in PfRh5 suggests that it alone may not be sufficient and blocking processing of a broad array of protein substrates processed by PMX is important to explain the observed potency of these compounds in inhibiting growth of *P. falciparum*¹¹. WM382 has been shown to be a potent inhibitor of *P. falciparum* at liver, blood and transmission stages and further development is in progress of a clinical candidate that can be tested in human clinical trials as an antimalarial drug.’

Reviewer #2 (Remarks to the Author):

Review of paper by T. Triglia et al. submitted to Nature Medicine. This is an outstanding paper that describes the processing of RH5 by PMX. There is a great deal of detail and methods that increase our understanding of the RH5 complex in invasion. They show that the binding to Basigin requires the cleaving of the N-terminus of RH5. They produced cleaved and uncleaved RH5 and show that only cleaved will bind to Basigin. I would point out that the discovery of RH5 and Basigin (ref 22 Crosnier C. et al) identified the binding with intact RH5 and this worked because they used the AVEXIS system that allows identity although the interaction is low affinity.

Authors response: We thank Dr Miller for raising this very good point and it is one we should have included in the discussion – we have now done so.

As explanation the full-length RH5 was expressed in the AVEXIS system by Crosnier et al and they showed that it apparently bound basigin. In this system the RH5 protein is expressed as a fusion. However, a subsequent publication (Crosnier et al, 2013; Mol Cell Proteomics 12, 3976), shows that the full-length RH5 with a ~25kDa C-terminal rat Cd4 tag, migrates at less than 70 kDa, suggesting that RH5 was processed in the HEK293 cells used to ~45kDa. We and others also find that endogenous mammalian proteases cleave PfRh5 at a position close to where Plasmepsin X would cleave in *P.falciparum*. This suggests that the original Crosnier et al publication showing Rh5-basigin binding, was a processed PfRh5 that had the 15 kDa Prodomain removed. This is consistent with the western blots shown in both Crosnier et al manuscripts mentioned above.

A paragraph has now been added in discussion as follows:

‘Previously, PfRh5 has been expressed in mammalian cells as a full-length fusion protein in a pentameric platform and this enabled the detection and identification of basigin as the receptor to which it bound on the erythrocyte surface²². This result would be inconsistent with our demonstration that full-length PfRh5 did not bind basigin because the 15 kDa prodomain blocked the binding site. However, a subsequent publication showed that the PfRh5 protein with a ~25 kDa Cd4 tag, migrates at 70 kDa suggesting that PfRh5 had been processed. The PfRh5 prodomain that is removed by PMX is predicted to be highly disordered and readily cleaved by unknown proteases both in insect³⁶, Schneider 2³⁷ and in our hands mammalian heterologous expression systems. Therefore, assays testing the ability of PfRh5 to bind

erythrocytes and basigin directly would have very likely used the processed PfRh5 that lacked the N-terminal domain consistent with our results that the N-terminal prodomain blocks basigin binding.’

My one concern is Fig. 6 in which they introduce a new term, prenome, a subclass of micronemes. Evidence for this is not mentioned in the results section. This organelle enters the rhoptry apical end. There is no data to substantiate the process that they describe. The minimum would be electron microscopy showing these events. Without these data, they need to say that this division of micronemes is purely speculative. More important, they need to develop these data in the results or leave it for another publication. I am recommending acceptance of the paper when corrected for Nature Medicine.

Authors response: We thank Dr Miller for these comments and agree we have not used immuno-EM so cannot definitively prove the existence of prenemes and postnemes. Consequently, we have removed these names from the figure and discussion.

The model we show in Fig. 6 is a hypothesis that we present in the discussion only and not in the results section. This model is based on our biochemical data, and it provides evidence there are more than one type of microneme and there is already data that supports this from previous studies in both Plasmodium and Toxoplasma which we have quoted in the manuscript. We have reworded this discussion to show it is clearly a hypothesis based on our biochemical data and quoted these previous studies.

‘The *P. falciparum* egress programme begins around 9 min before merozoites exit the erythrocyte (Fig. 6a). Both PMX and SUB1 reside in the exonemes⁹, and following their fusion and discharge, SUB1, released into the parasitophorous vacuole space, begins the process of parasitophorous vacuole breakdown (Fig. 6b). Micronemes containing EBA and AMA1 proteins fuse at the rhoptry neck before invasion but SUB2, which also has a microneme subcellular localisation, is discharged on the surface of the merozoite proteins during active invasion³¹. This suggests that there are different forms of micronemes as has been suggested previously for both *P. falciparum*^{16, 32, 33} and *Toxoplasma gondii*³⁴. Our data adds support to this idea and results from fractionation of C1-treated merozoites, supports the idea that there are different types of micronemes that contain distinct cargo (Fig. 6a-e). We propose that PCRCR, EBAs, PFRhs, AMA1 and PMX are stored in micronemes type 1 and 2 before egress. Consistent with this is the observation that PMX processing of micronemal contents does not result in full processing of all proteins and increases markedly when microneme fusion was blocked by C1 (Fig. 6f). This suggests that these proteins are separated from PMX at some stages and that either there are likely two early microneme subsets, one of which contains PMX (Fig. 6g), or alternatively these micronemes have subcompartments (Fig. 6h). Previous evidence for microneme subsets in *P. falciparum*^{16, 32, 33} have suggested that EBA175 and AMA1 are in separate micronemes. The proposed microneme subsets are in addition to the exonemes¹⁵ and mononemes³⁵ previously defined. The compartmentalisation of proteins destined for processing by PMX and relocation to the merozoite surface during invasion provides a mechanism to time specific processing events and release of protein as required for the finely controlled process of merozoite invasion.’

Reviewer #3 (Remarks to the Author):

A complex of proteins including Pfrh5, which the authors have recently identified new members for and renamed the whole complex PCRCR, plays a key role in the invasion of human erythrocytes by Plasmodium falciparum merozoites. Several members of this complex are proteolytically cleaved during the invasion process, as are numerous other P. falciparum invasion proteins. In this manuscript, the authors use a combination of CRISPR-Cas9 engineering and protease inhibitors to explore the role that this processing plays for PCRCR location and function. The manuscript is well written, clear, contains extensive data and will undoubtedly add significantly to our understanding of this critical step during invasion. It should also be noted that in several places the authors are absolutely pushing the boundaries of technology – Figure 2 in particular is a genetic tour de force, this combination of inducible knockouts and complementation with different variants is rarely performed in P. falciparum and to be applauded. The only significant concern I have is that the hypotheses about different kinds of secretory organelles (prenemes/postnemes etc) and their roles in invasion veers a little far into the speculative, and the figures presenting them lend an air of definitiveness that is not currently backed up by the data. Other than that, a few relatively minor technical items should be addressed in a revised manuscript.

General Authors response: We thank the reviewer for taking the time to read the paper and providing excellent feedback.

Major issues

1) Strain adaptation. The authors note that two of the Pfrh5 PMX cleavage-site edited lines, NFAA and NALA, took significantly longer to grow following transfection (lines 122-124), which matches with the more significant Pfrh5 processing defect in these lines. It would be helpful to provide more details here – how much longer did they take to come back, and was there a persistent growth defect, or once the parasites were recovered, did they grow at the same rate as any other line? A formal growth rate comparison post-recovery would be helpful. However, the deeper concern with all P. falciparum transfections is that during the extended time it takes to recover lines, other genetic/epigenetic changes have emerged in addition to the targeted mutation. These could be stochastic but could also be compensatory – ie. other changes were needed to compensate for these specific PMX site mutations. RNAseq and/or whole genome sequencing of the edited lines would help provide reassurance that the phenotypes observed are not in part due to background changes in addition to the targeted mutations/or that additional changes were needed for the NFAA and NALA mutations to be viable. Of course, this is a common feature of all P. falciparum transfections, and is almost always overlooked, so it is acknowledged that this is asking the authors to go further than almost all similar papers do. There may be good reasons why this is not required – it would be helpful though to hear that reasoning articulated.

Authors response: We thank the reviewer for these comments and we have gone back over this data and we believe that the differences in transfection time were overstated. These transfectants represent one transfection each and they did not show any apparent differences in growth rate to wild type or other transfectants over the time they were analysed including at the beginning. We frequently get distinct differences in the time that particular constructs grow to a detectable parasitemia.

For the reviewer we have assembled the times that each transfectant took to grow from the initial transfection to 1% parasitemia. As you can see the difference between the wt NFLQ transfectant and the other mutant lines is 4 days at most and usually less. This is unlikely to be a significant difference over that time and unlikely that any of these differences are significant.

We thank the reviewer for raising this as it made us go back over this data and realise it was not a significant difference and enabled us to correct our statement.

Transfection	Days for transfectant to appear	Days for 1% parasitaemia
NFLQ	9	18
NFLA	9	19
AFLA	9	22
NFAA	12	20
NALA	9	20

Therefore we have removed the sentence:

‘Interestingly, the NFAA and NALA mutant transfectants took significantly longer to grow following transfection suggesting they had a major growth disadvantage.’

2) Function of interacting proteins. Figure 3 is a tour de force, and almost a manuscript in itself – pull down of interacting proteins, then generating inducible knockouts of the encoding genes to assess function, is a huge amount of work. The implications for PCRCR function/PfRh5 function however are not completely clear. The authors test for an effect on PfRh5 processing in the iKO lines and see no effect, but what about PfRh5 trafficking and/or PCRCR complex formation? If none of these elements are affected, then is the inference that the interactions detected by co-immunoprecipitation are actually false positives?

Authors response: Yes we agree with the reviewer and have made this point clearer and use the words false-positive in both the results and discussion to make it clear as below. Despite the recognition that these proteins are false positives we believe it is an important part of the story within the manuscript. Also it provides evidence that P113 plays no role with the PCRCR function in support of other studies.

Results: ‘Demonstration that P113 was not required for *P. falciparum* merozoite invasion or growth was consistent with it not playing a role in the function of PfRh5²⁶ and that the other proteins detected in these pull downs with PfRh5 were false positives.’

Discussion: ‘The presence of a highly disordered prodomain at the N-terminus of PfRh5 that can be processed by different proteases in heterologous systems also suggest it can interact with different proteins. The identification of P113, GRP170, Sel1, and 10TM is consistent with this, however, the analysis of conditional knockouts of the corresponding genes encoding these proteins showed that they have no effect on PMX processing of PfRh5 suggesting that these interactions are not functionally relevant and false positives.’

3) Trafficking models. Figure 6 lays out an extended model of merozoite invasive organelles, suggesting there are two different kinds of micronemes (prenemes and postnemes) and that prenemes are further functionally divided. While developing new hypotheses to bring the field forward is of course highly commendable and useful, the fact remains that the definition of these new organelles/subtypes of micronemes does not include direct visualisation/microscopy – it is all theoretical, based on the authors’ interpretation of inhibitor experiments, and is therefore highly speculative. The concern is that these terms and definitions, laid out in a very definitive way in Figure 6, will enter standard usage without solid experimental justification.

Discussing the potential implications of the processing data for trafficking is reasonable enough, but Figure 6 is, in my opinion, advancing hypothesis too far into speculation.

Authors response: We thank the reviewer for these comments and agree we cannot definitively prove the existence of prenexemes and postnexemes and stated it was a hypothesis and was in the Discussion. Consequently, we have removed these names from the figure and discussion.

The model we show in Fig. 6 is a hypothesis that we present in the discussion only and not in the results section. This model is based on our biochemical data, and it provides evidence there are more than one type of microneme and there is already data that supports this from previous studies in both *Plasmodium* and *Toxoplasma*. We have reworded this discussion to show it is clearly a hypothesis based on our biochemical data and quoted these previous studies.

‘The *P. falciparum* egress programme begins around 9 min before merozoites exit the erythrocyte (Fig. 6a). Both PMX and SUB1 reside in the exonemes⁹, and following their fusion and discharge, SUB1, released into the parasitophorous vacuole space, begins the process of parasitophorous vacuole breakdown (Fig. 6b). Micronemes containing EBA and AMA1 proteins fuse at the rhoptry neck before invasion but SUB2, which also has a microneme subcellular localisation, is discharged on the surface of the merozoite proteins during active invasion³¹. This supports the hypothesis that there are different forms of micronemes as has been suggested previously for both *P. falciparum*^{16, 32, 33} and *Toxoplasma gondii*³⁴. Our data adds support to this idea and results from fractionation of C1-treated merozoites suggests there are different types of micronemes that contain distinct cargo (Fig. 6a-e). We hypothesise that PCR, EBAs, PfRh5, AMA1 and PMX are stored in micronemes type 1 and 2 before egress. Consistent with this is the observation that PMX processing of micronemal contents does not result in full processing of all proteins and increases markedly when microneme fusion was blocked by C1 (Fig. 6f). This suggests that these proteins are separated from PMX at some stages and that either there are likely two early microneme subsets, one of which contains PMX (Fig. 6g), or alternatively these micronemes have subcompartments (Fig. 6h). Previous evidence for microneme subsets in *P. falciparum*^{16, 32, 33} have suggested that EBA175 and AMA1 are in separate micronemes. The proposed microneme subsets are in addition to the exonemes¹⁵ and mononemes³⁵ previously defined. The compartmentalisation of proteins destined for processing by PMX and relocation to the merozoite surface during invasion provides a mechanism to time specific processing events and release of protein as required for the finely controlled process of merozoite invasion.’

Minor items

1) Definitiveness of interpretation of immunoblots. Much of the processing data interpretation depends on the presence/absence of bands on immunoblots, where the authors will know well different exposure times can sometimes support very different interpretations. The blots are clear and well controlled, so this is not a significant concern, but statements like “NFAA and NALA mutants were not processed at the NFLQ site” (line 133) is more absolute than Fig1c, with a faint processed band that could perhaps be lighter or darker depending on exposure, actually warrants. A simple tempering of language, and acknowledging where appropriate that it is not possible to be absolute based on immunoblot data alone is all that is required.

Authors response: We thank the reviewer for these comments and have both reworded the sentence as below and modified Fig. 1d to reflect the trace cleavage at the NFLQ site in the NFAA and NALA mutant parasites:

‘While the NFLA and AFLA mutant parasites showed some cleavage at the NFLQ site, both NFAA and NALA mutants showed only trace amounts of cleavage at the NFLQ site (Fig. 1c).’

New Fig. 1d

Mutated residues	Mutated site	p50 (NFLQ) Cleavage
P2’	NFLA	Yes (partial)
P2 + P2’	AFLA	Yes (partial)
P1’ + P2’	NFAA	Trace
P1 + P2’	NALA	Trace

2) Quantification of co-localisation. The PfRh5 prodomain is stated to co-localise with full-length PfRh5, and Ext Fig 3 is compelling, but only two schizonts are shown for each co-localisation. How many schizonts were measured overall using super-resolution microscopy to make such an absolute statement about co-localisation? Being less absolute, and more quantitative (eg Rh5HA/Rh5nGreen co-localise x% +/- y% in n schizonts, and Rh5HA/Rh5delPDnGreen co-localise a% +/- b% in n schizonts) would be more accurate and provide more reassurance in the data.

Authors response: We thank the reviewer for pointing this out and an additional panel has been added to the Extended Figure 3 as panel c. The number of schizonts analysed along with the additional information is contained in the Figure legend as below. Source data will also be included if the manuscript is accepted.

‘Extended Data Fig. 3: The Rh5PD has no role in PfRh5 trafficking. **a,** The Rh5PD is not required for post-Golgi trafficking to an apical organelle. Super-resolution images showing colocalization of Rh5HA and Rh5nGreen in late schizonts. Intensity plots along the white broken line show colocalization in the apical organelles. Scale bar 2 μ m. **b,** Super-resolution images showing colocalization of Rh5HA and Rh5 Δ PDnGreen in late schizonts. Intensity plots along the white broken line show colocalization in the apical organelles. Scale bar 2 μ m. **c,** Pearson’s colocalization index determined for 10 schizonts of each transfectant. Mean values are shown together with error bars representing standard deviation. Students t-test was used to calculate the p-value. Source data are provided for this Figure.’

REVIEWERS' COMMENTS

Reviewer #3 (Remarks to the Author):

No further comments - thanks to the authors for so rapidly and comprehensively addressing the questions from all reviewers. It's a really great manuscript - they should be very proud of the rigour and scope of the work.